# Optimistic Regret Bounds for Online Learning in Adversarial Markov Decision Processes

**Sang Bin Moon**[1]  **Abolfazl Hashemi**[1]

[1]School of Electrical and Computer Engineering, Purdue University, West Lafayette, Indiana, USA

## Abstract

The Adversarial Markov Decision Process (AMDP) is a learning framework that deals with unknown and varying tasks in decision-making applications like robotics and recommendation systems. A major limitation of the AMDP formalism, however, is pessimistic regret analysis results in the sense that although the cost function can change from one episode to the next, the evolution in many settings is not adversarial. To address this, we introduce and study a new variant of AMDP, which aims to minimize regret while utilizing a set of cost predictors. For this setting, we develop a new policy search method that achieves a sublinear optimistic regret with high probability, that is a regret bound which gracefully degrades with the estimation power of the cost predictors. Establishing such optimistic regret bounds is nontrivial given that (i) as we demonstrate, the existing importance-weighted cost estimators cannot establish optimistic bounds, and (ii) the feedback model of AMDP is different (and more realistic) than the existing optimistic online learning works. Our result, in particular, hinges upon developing a novel optimistically biased cost estimator that leverages cost predictors and enables a high-probability regret analysis without imposing restrictive assumptions. We further discuss practical extensions of the proposed scheme and demonstrate its efficacy numerically.

## 1 INTRODUCTION

Reinforcement learning studies the problem of sequential decision-making modeled as a Markov Decision Process (MDP), where a learner interacts with an environment and solves the optimal policy that minimizes the cumulative cost incurred by the environment. The learner interacts with the environment by observing a state, choosing an action, and suffering a cost, repeatedly for a finite number of time steps. The process is sequential in the sense that the chosen action affects the environment state, and thus the next state is observed through a stochastic transition probability function, and the cost suffered by the learner is determined by an unknown cost function accordingly. After a number of episodes, one can measure the performance of the learner's policy with regret, i.e., how larger the total cost suffered by the learner is compared to the total cost of a fixed optimal policy in hindsight. MDPs are useful for decision-making in various fields, such as robotics [Akkaya et al., 2019], finance [Wei et al., 2019, Buehler et al., 2019], and healthcare [Tsoukalas et al., 2015]. However, in many real-world applications, the tasks and environment may change over time, leading to non-stationary dynamics. In such cases, the assumptions of MDP may not hold, and the performance of the decision-making system may deteriorate.

In this paper, we consider the problem of learning policies in Adversarial MDP (AMDP) as a generalization of the traditional MDP model, where the environment can choose different cost functions for each episode. AMDP gives greater flexibility to account for changing environments and even the existence of other agents. For example, AMDP can model an energy-efficient drone navigation problem [Hong et al., 2021], where wind incurs higher energy consumption while it is not observed in advance and changes arbitrarily. Stochastic inventory control [Even-Dar et al., 2009] can also be modeled as AMDP, because item price and inventory cost change from time to time due to economic conditions. Eventually, AMDP can be extended to hierarchical or multi-agent problems, because parent policy or other agents evolve and incur different costs to a learner. Existing online learning [Even-Dar et al., 2009, Yu et al., 2009, Zimin and Neu, 2013, Neu et al., 2010a,b, 2014, Jin et al., 2020] and policy optimization approaches [Shani et al., 2020, Luo et al., 2021] to AMDP solves the optimization problem to minimize the cost in hindsight. However, it can be too restrictive and result

in conservative regret bounds. For instance, in multiplayer games, the action, and in turn the policies of other players may be predicted from simulation and historical observation; this insight if leveraged properly may lead to turning the game to a specific player's advantage [Vundurthy et al., 2023].

Motivated by this shortcoming, we propose to study a new formulation for RL with time-varying cost functions where the aim is to learn a policy that minimizes its regret while resorting to a given set of time-varying predictive estimators of the cost functions, denoted by $\{c_t\}_{t=1}^T$ and $\{M_t\}_{t=1}^T$, respectively. We propose a novel policy search scheme that utilizes the set of optimistic cost predictors and achieves sublinear regret bounds. Specifically, we make the following contributions:

- We show the worst-case regret bound of $\tilde{\mathcal{O}}\left(\sqrt{d\left(\{c_t\}_{t=1}^T, \{M_t\}_{t=1}^T\right)}\right)$ for the full-information feedback setting[1] and $\tilde{\mathcal{O}}\left(d(\{c_t\}_{t=1}^T, \{M_t\}_{t=1}^T)^{2/3}\right)$ in expectation for bandit feedback setting, where $d(\cdot, \cdot)$ captures cumulative estimation error of the cost predictors. It is also shown that with high probability the algorithm achieves the regret bound of $\tilde{\mathcal{O}}\left(d(\{c_t\}_{t=1}^T, \{M_t\}_{t=1}^T)^{3/4}\right)$. These regret bounds are optimistic in nature, i.e., the bound scales with the prediction power of optimistic cost predictors, and can lead to constant regret with perfect prediction. In the worst case, on the other hand, the proposed scheme to learn a policy satisfies sublinear regret bounds.

- Crucial to the establishment of these results is the development of a new cost estimator. This new estimator leverages the bandit information about the cost as well as the set of predictive estimators to update the policy. We show the proposed estimator has variance-reduction benefits and thus it may be of independent interest in similar problems.

- We also introduce the anytime extensions for continuous training beyond the fixed number of episodes and establish similar regret guarantees. Then we generalize the setting to the unknown transition setting and establish high probability regret bounds by leveraging the idea of transition estimation via confidence sets.

## 2 BACKGROUND AND RELATED WORK

We start with the precise definition of an AMDP. A standard definition follows an episodic loop-free AMDP [Zimin and Neu, 2013] or a loop-free stochastic shortest path [Neu et al., 2012].

**Definition 1.** *An episodic loop-free Adversarial Markov decision process (AMDP) is a tuple*

---
[1]Recall the notation $\tilde{\mathcal{O}}(\cdot)$ hides the logarithmic terms in its argument.

$\mathcal{M} = (\mathcal{X}, \mathcal{A}, \mathbb{P}, L, \{c_t\}_{t=1}^T)$ *which consists of a finite discrete state space denoted by $\mathcal{X}$, a finite discrete action space denoted by $\mathcal{A}$, a probabilistic transition function denoted by $\mathbb{P} : \mathcal{X} \times \mathcal{A} \times \mathcal{X} \to [0,1]$, and a sequence of cost functions denoted by $c_t : \mathcal{X} \times \mathcal{A} \to \mathbb{R}$ such that:*

- *The cost functions are bounded, that is, $c_t \in [0,1]^{|\mathcal{X}| \times |\mathcal{A}|}$ for $t = 1, 2, \ldots, T$.*

- *The state space $\mathcal{X}$ is partitioned into $L$ non-overlapping layers $\mathcal{X}_0, \mathcal{X}_1, \ldots, \mathcal{X}_L$ such that $\mathcal{X} = \cup_{l=0}^L \mathcal{X}_l$ and, it holds that $\mathcal{X}_{l_1} \cap \mathcal{X}_{l_2} = \emptyset$ for any $l_1 \neq l_2$.*

- *The state transition function $\Pr(x'|x,a)$ is stationary.*

- *If for some $x \in \mathcal{X}_l$ and some layer $l \in \{0, \ldots, L-1\}$, $\Pr(x'|x,a) > 0$, then $x' \in \mathcal{X}_{l+1}$; that is, state transition happens only between two consecutive layers.*

- *$\mathcal{X}_0$ and $\mathcal{X}_L$ are singletons; that is, $\mathcal{X}_0 = \{x_0\}$ and $\mathcal{X}_L = \{x_L\}$.*

**Policy search in AMDP.** Online learning approaches to MDP, such as Follow-the-Regularized-Leader (FTRL) or Online Mirror Descent (OMD), solve the linear optimization problem with occupancy measure $\rho$. Occupancy measure quantifies the joint probability of the probability of visiting a state $x$ and the probability of taking an action $a$ given the state. Thus, conversely, an occupancy measure controls the behavior of an agent under a stationary, stochastic, and known/unknown transition probability distribution. The behavior is governed by the policy $\pi$ defined as

$$\pi_t(a|x) = \frac{\rho_t(x,a)}{\sum_{a' \in \mathcal{A}} \rho_t(x,a')}. \quad (1)$$

Therefore, given an MDP, the optimization objective is to minimize the total cost suffered by an occupancy measure. Since occupancy measure quantifies the probability of a specific state and action pair, the total (expected) cost can be formulated by a linear objective function with respect to a cost function $c$, i.e., $\langle \rho_t, c_t \rangle = \sum_{x \in \mathcal{X}, a \in \mathcal{A}} c_t(x,a) \rho_t(x,a)$. This leads to the following definition of regret (w.r.t. the policy corresponding to $\rho$) that underlies the problem of learning policies in AMDPs,

$$\mathcal{R}_T(\rho^*, \{c_t\}_{t=1}^T) = \sum_{t=1}^T \langle \rho_t - \rho^*, c_t \rangle. \quad (2)$$

Here, $\rho \in \Delta(\mathcal{M})$ where $\Delta(\mathcal{M})$ denote the space of all occupancy measures over AMDP $\mathcal{M}$, $\langle ., . \rangle$ represents the Euclidean inner product over the space of $\mathcal{X} \times \mathcal{A}$, and $\rho_t$ denotes the agent's selected occupancy measure in episode $t$.

OREPS [Zimin and Neu, 2013] is the baseline algorithm for learning policies in AMDPs that solves the constrained, regularized regret minimization problem via

a mirror descent update with stepsize $\eta$, i.e., $\rho_{t+1} = \arg\min_{\rho \in \Delta(\mathcal{M})} \eta \langle \rho, c_t \rangle + D_R(\rho \| \rho_t)$, where $R$ is negative entropy

$$R(\rho) = \sum_{x \in \mathcal{X}, a \in \mathcal{A}} \rho(x, a) \log \rho(x, a) - \sum_{x \in \mathcal{X}, a \in \mathcal{A}} \rho(x, a),$$

and $D_R$ is the unnormalized KL divergence being the corresponding Bregman divergence [Abernethy and Rakhlin, 2009, Lattimore and Szepesvári, 2018]

$$D_R(\rho \| \rho') = \sum_{x \in \mathcal{X}, a \in \mathcal{A}} \rho(x, a) \log \frac{\rho(x, a)}{\rho'(x, a)}$$
$$- \sum_{x \in \mathcal{X}, a \in \mathcal{A}} (\rho(x, a) - \rho'(x, a)). \tag{3}$$

KL divergence regularizes the information loss from the history that previous solutions were optimized for. OREPS solves the unconstrained version of the original problem and the dual formulation of the projection onto $\Delta(\mathcal{M})$.

**Optimistic online learning.** Let $\{M_t\}_{t=1}^T$ be a sequence of time varying predictive estimators such that $M_t : \mathcal{X} \times \mathcal{A} \to [0, 1]$ for all $t$. For online linear optimization, Rakhlin and Sridharan [2013] show that optimistic mirror descent (OMD) [Chiang et al., 2012] equipped with a similar cost predictor sequence can achieve optimistic regret bounds, i.e., $\tilde{\mathcal{O}}(\sqrt{d(\{c_t\}_{t=1}^T, \{M_t\}_{t=1}^T)})$, where $d(\cdot, \cdot)$ captures cumulative estimation error of the cost predictors. This result shows with perfect estimation the regret is $\tilde{\mathcal{O}}(1)$ while for futile estimation, i.e., the worst case, the regret is $\tilde{\mathcal{O}}(\sqrt{T})$. In this paper, we aim to establish optimistic regret bounds for a class of policy search methods in AMDPs. In contrast to Rakhlin and Sridharan [2013], our setting is more general in the sense that it accounts for the dynamic and state-full nature of the interaction between the learner and the environment which is captured by the notion of state space. Further, although Rakhlin and Sridharan [2013] leverages the method from Abernethy et al. [2012] to propose a no-regret scheme for the bandit setting in online linear optimization, their algorithm is not applicable in our setting since the bandit feedback model of the present paper is different from Rakhlin and Sridharan [2013] and more meaningful in the sense that the learner observes the cost of the chosen action, not the mixture of cost of all feasible actions. Consequently, the proposed method and its analysis differ considerably from Rakhlin and Sridharan [2013]. Further, we leverage a single-projection method adopted from Joulani et al. [2017] to reduce the computational cost of optimistic policy search compared to OMD which requires two projection steps.

**Bandit cost estimation.** Learning a policy in the bandit case relies on estimating the unknown cost function for each episode. Given the connection of AMDPs to adversarial bandits, Zimin and Neu [2013] incorporate the celebrated importance-weighted cost estimator in OREPS which was originally exhibited in the EXP3 algorithm [Cesa-Bianchi and Lugosi, 2006]. Recently, Jin et al. [2020], Ghasemi et al. [2021] have utilized the implicit exploration estimator from Neu [2015], i.e.,

$$\hat{c}'_t(x, a) = \frac{c_t(x, a)}{\rho_t(x, a) + \gamma} \mathbb{I}\{(x, a) \in \bar{\mathbf{u}}_L(t)\}, \tag{4}$$

in a similar OREPS-based update, where $\gamma \geq 0$ is the exploration parameter and $\bar{\mathbf{u}}_L(t)$ denotes the history of states and actions up to and including the $L^{\text{th}}$ layer of episode $t$. As we discuss later, such estimators fail to result in optimistic regret guarantees that degrade gracefully with $d(\{c_t\}_{t=1}^T, \{M_t\}_{t=1}^T)$. Thus, we develop a new cost estimator, characterize its properties, and show that it results in optimistic bounds.

# 3 OPTIMISTIC LEARNING IN AMDPS

Given that in the bandit setting, we need to resort to cost estimation, the estimation error of the estimator is an integral part of the regrets of the underlying algorithms. In order to establish optimistic bounds, our regret analysis shows that it is crucial to have an estimator whose error is controlled with $d(\{c_t\}_{t=1}^T, \{M_t\}_{t=1}^T)$. Let us consider the estimator (4), define $\mathbb{E}_t[\cdot] = \mathbb{E}[\cdot | \mathbf{u}(t)]$, and examine $\mathbb{E}_{t-1} \| \hat{c}'_t - M_t \|^2$ which can be thought of as some notion of variance. Note that (4) with $\gamma = 0$ may suffer from an unbounded variance.[2] With $\gamma > 0$ immediate calculation shows $\mathbb{E}_{t-1}[(\hat{c}'_t(x, a) - M_t(x, a))^2]$ cannot be written as a function of $|c_t(x, a) - M_t(x, a)|$ which, as our regret analysis demonstrates, results in failure of achieving optimistic expected regret bounds when utilizing (4) with $\gamma \geq 0$.

We thus propose a new cost estimator that provably results in an optimistic expected regret bound in conjunction with a mirror descent-based update. The proposed estimator defined for all $\gamma \geq 0$ is as follows

$$\hat{c}_t(x, a) \tag{5}$$
$$= \frac{c_t(x, a) - M_t(x, a)}{\rho_t(x, a) + \gamma} \mathbb{I}\{(x, a) \in \bar{\mathbf{u}}_L(t)\} + M_t(x, a).$$

Crucially, the proposed estimator $\hat{c}_t(x, a)$ leverages the predictive estimators $M_t(x, a)$. In particular, in contrast to (4) the unexplored state and action pairs incur the cost predicted by $M_t(x, a)$ as opposed to incurring zero cost. Also, Wei and Luo [2018] suggested a similar cost estimator as (5) with $\gamma = 0$ for the multi-armed bandit problem. However, our estimators in this paper address the problem of learning in MDPs and exploration parameter $\gamma > 0$ is crucial to

---

[2]This property is known to be the underlying reason that EXP3 cannot satisfy sublinear regret with high probability in adversarial bandits [Lattimore and Szepesvári, 2018].

our analysis of high probability guarantee with Lemma 2 in Appendix A.4.

Lemma 1 studies the statistical properties of the proposed estimator.

**Lemma 1.** *The proposed cost estimator* (5) *satisfies*

$$\mathbb{E}_{t-1}[\hat{\boldsymbol{c}}_t(x,a)] = \frac{\rho_t(x,a)c_t(x,a) + \gamma M_t(x,a)}{\rho_t(x,a) + \gamma},$$

$$\mathbb{E}_{t-1}\big[(\hat{\boldsymbol{c}}_t(x,a) - M_t(x,a))^2\big] \le \frac{(c_t(x,a) - M_t(x,a))^2}{\rho_t(x,a) + \gamma}.$$

**Variance reduction property.** This result shows that if $\gamma > 0$ the variance is provably bounded. Furthermore, if $M_t(x,a) \le 2c_t(x,a)$ for all $(x,a) \in \mathcal{X} \times \mathcal{A}$ and $t = 1,\ldots T$, immediate calculation shows $|c_t(x,a) - M_t(x,a)|^2 \le |c_t(x,a)|^2$. That is, the proposed estimator enjoys a lower variance compared to (4). Also if the predictors $\{M_t\}_{t=1}^T$ are optimistic, i.e., $M_t(x,a) \le c_t(x,a)$, for all $t = 1,\ldots,T$ and $(x,a) \in \mathcal{X} \times \mathcal{A}$ then the proposed cost estimator (5) is an *optimistically biased* estimator given that

$$\mathbb{E}_{t-1}[\hat{\boldsymbol{c}}_t(x,a)] = \frac{\rho_t(x,a)c_t(x,a) + \gamma M_t(x,a)}{\rho_t(x,a) + \gamma} \le c_t(x,a).$$

Therefore, as long as $M_t(x,a) \le c_t(x,a)$, compared to (4), the proposed estimator has the same bias while having a lower variance. Note that the condition $M_t(x,a) \le c_t(x,a)$ is very mild and may be ensured in a variety of non-adversarial settings based on the observed cost signal. Finally, note that different from (4) the variance of the proposed estimator is controlled by the estimation power of the cost predictors. A feature we will leverage to achieve optimistic regret bounds.

With the proposed cost estimator, we then utilize it in a mirror-descent type update by adopting the result of Joulani et al. [2017]. In particular, given $\rho_t$ the agent runs an episode exploration subroutine and subsequently employs

$$\rho_{t+1} = \arg\min_{\rho \in \Delta(\mathcal{M})} \eta\langle\rho, \hat{\boldsymbol{c}}_t + M_{t+1} - M_t\rangle + D_R(\rho\|\rho_t). \quad (6)$$

Please see Algorithm 1 for a detailed description of the learning process. We call the resulting scheme OREPS-OPIX. Analogous to the standard MD and OREPS algorithms, this update can be tackled efficiently through a well-known two-step procedure [Abernethy and Rakhlin, 2009, Lattimore and Szepesvári, 2018, Zimin and Neu, 2013]. Specifically, by adopting the result of Zimin and Neu [2013],

$$\rho_{t+1}(x,a) = \frac{\rho_t(x,a)e^{\beta(x,a|\hat{v}_t,\hat{\boldsymbol{c}}_t)}}{\sum_{x' \in \mathcal{X}_l, a \in \mathcal{A}} \rho_t(x',a)e^{\beta(x',a|\hat{v}_t,\hat{\boldsymbol{c}}_t)}}, \quad (7)$$

where $l$ denotes the layer in which state $x$ belongs to, $\beta$ is defined as

$$\beta(x,a|\hat{v}_t,\hat{\boldsymbol{c}}_t) = -\eta(\hat{\boldsymbol{c}}_t(x,a) + M_{t+1}(x,a) - M_t(x,a))$$
$$- \sum_{x' \in \mathcal{X}_{l+1}} \hat{v}_t(x')\Pr(x'|x,a) + \hat{v}_t(x),$$

---

**Algorithm 1** OREPS with Optimistic Predictor and Implicit eXploration (OREPS-OPIX)

**Require:** Learning rate $\eta$, exploration parameter $\gamma$
1: Initialize occupancy measure $\rho_1(x,a)$ as a uniform distribution over $x \in \mathcal{X}_l$ and $a \in \mathcal{A}$ for $l = 1, 2, \ldots, L-1$
2: Initialize cost predictor as $M_1 = 0$
3: **for** Episodes $t = 1, 2, \ldots, T$ **do**
4:     Initialize cost estimator as $\hat{\boldsymbol{c}}_t = 0$
5:     **for** Time steps $l = 1, 2, \ldots, L-1$ **do**
6:         Observe state $x_l \in \mathcal{X}_l$ from the environment
7:         Choose action $a_l \sim \rho_t(x_l, \cdot)$
8:         Observe cost $c_t(x_l, a_l)$
9:         Save $x_l, a_l$ and $c_t(x_l, a_l)$ to $u_t$
10:     **end for**
11:     **for** Tuples $x, a, c_t(x,a)$ in $u_t$ **do**
12:         $\hat{\boldsymbol{c}}_t(x,a) \leftarrow (c_t(x,a) - M_t(x,a))/(\rho_t(x,a) + \gamma) + M_t(x,a)$
13:         Update $M_{t+1}(x,a)$
14:     **end for**
15:     Solve $\rho_{t+1} = \text{argmin}_{\rho \in \Delta(\mathcal{M})} \eta\langle\rho, \hat{\boldsymbol{c}}_t + M_{t+1} - M_t\rangle + D_R(\rho\|\rho_t)$.
16: **end for**

---

and $\hat{v}_t$ is defined as

$$\hat{v}_t = \arg\min_v \sum_{l=0}^{L} \ln\left\{\sum_{x \in X_l, a \in A} \rho_t(x,a)e^{\beta(x,a|v,\hat{\boldsymbol{c}}_t)}\right\}.$$

Note that by setting $M_t = M_{t+1} = 0$, one recovers the OREPS algorithm. Further, in the full-information case, one can replace $\hat{\boldsymbol{c}}_t$ with the observed cost vector $c_t$.

## 4 OPTIMISTIC REGRET BOUNDS

In this section, we provide a detailed regret analysis of the proposed OREPS-OPIX scheme in (6) equipped with the proposed cost estimator in (5).

Theorem 1 establishes the regret bound under full information. For compactness, we denote the prediction error in episode $t$ as $\sigma_t = c_t - M_t$.

**Theorem 1** (Full information). *Under full information feedback, there exists a stepsize $\eta$ such that OREPS-OPIX satisfies*

$$\mathcal{R}_T(\rho^*, \{c_t\}_{t=1}^T) = \tilde{\mathcal{O}}\left(\sqrt{L\sum_{t=1}^T \|\sigma_t\|_\infty^2}\right). \quad (8)$$

To understand the benefit of leveraging cost predictors, assume $\sum_{t=1}^T \|c_t - M_t\|_\infty^2 = \mathcal{O}(T^\alpha)$ for some $0 \le \alpha \le 1$ where $\alpha = 0$ and $\alpha = 1$ correspond to perfect estimation and futile estimation, respectively. Then, if $\eta = \mathcal{O}(T^{-\alpha/2})$,

we have $\mathcal{R}_T(\rho^*, \{c_t\}_{t=1}^T) = \tilde{\mathcal{O}}(T^{\alpha/2})$. That is, the regret can be constant while in the worst case, the regret is $\tilde{\mathcal{O}}(\sqrt{T})$.

A downside of Theorem 1 is the requirement of full information on $c_t \in [0,1]^{|\mathcal{X}| \times |\mathcal{A}|}$ which is not a realistic assumption. Therefore, we next establish a bound on the expected regret of OREPS-OPIX under bandit feedback. As we discussed before, establishing optimistic regret bounds in the bandit setting for AMDPs seems to necessitate utilizing an estimator with bounded variance. Following Neu et al. [2010a], one could impose an assumption that ensures $\rho_t(x, a) > \alpha$ and establish regret bounds that scales with $\mathcal{O}(\alpha^{-1})$. Instead, we set $\gamma > 0$ but impose the mild assumption that the cost predictors $\{M_t\}_{t=1}^T$ are *optimistic*, i.e., $M_t(x, a) \le c_t(x, a)$.

Fei et al. [2020] proposed an algorithm that directly estimates a state-action value function instead of a cost function that is used to exponentially update a policy. They further extended the algorithm to alternately update policy and value function twice, mirroring the two-step optimization of OMD. Conceptually, it is analogous to having a predictor as a Q-function that is updated with the previous episode's cost function. In the worst case, their static regret bound, where $P_T = 0$, scales as $O(\sqrt{T})$. Zhao et al. [2022] investigated ensemble algorithms and imposed a lower bound on the occupancy measure for all states and actions. This regularization serves to bound the difference between the losses incurred by any two policies. They also explored optimistic variants by incorporating the two-projection OMD as originally proposed by Rakhlin and Sridharan [2013], Chiang et al. [2012]. By leveraging this optimistic algorithm, they achieve static regret bounds of $\tilde{\mathcal{O}}(L\sqrt{\sum^T \|c_t - M_t\|_\infty^2})$ in expectation, as opposed to 8. It is worth noting that both works exclusively explored the full information setting. In the subsequent discussion, we analyze the bandit feedback setting.

**Theorem 2** (Bandit – Expected). *Under bandit feedback, there exists a stepsize $\eta$ and an exploration parameter $\gamma$ such that OREPS-OPIX utilizing the proposed cost estimator* (5) *satisfies*

$$
\begin{aligned}
&\mathbb{E}[\mathcal{R}_T(\rho^*, \{c_t\}_{t=1}^T)] \\
&= \tilde{\mathcal{O}}\left( L^{\frac{1}{3}} \left( \sum_{t=1}^T \|\sigma_t\|_2^2 + \|\sigma_t\|_1 \right)^{\frac{2}{3}} \right). \quad (9)
\end{aligned}
$$

Note that the regret bound is optimistic as it scales with the estimation power of the cost predictors. Further, leveraging cost predictors is beneficial in the bandit feedback setting. In particular, the result of Theorem 2 demonstrates if $\sum_{t=1}^T \|c_t - P_t\|_1 = \mathcal{O}(T^{\alpha-1})$ for some $0 \le \alpha \le 1$ setting $\eta = \mathcal{O}(T^{-2\alpha/3})$ and $\gamma = \mathcal{O}(T^{-\alpha/3})$, OREPS-OPIX with the proposed cost estimator suffers $\tilde{\mathcal{O}}(T^{2\alpha/3})$ worst-case expected regret. Therefore, in the best case, the expected regret

is constant while in the worst case, the regret is $\tilde{\mathcal{O}}(T^{2/3})$. Note that here our theoretical results may be sub-optimal in the worst-case as we cannot achieve $\tilde{\mathcal{O}}(\sqrt{T})$ worst-case expected regret. Further study in this direction is a valuable future work.

Also, Wei and Luo [2021] achieved the dynamic regret bound of $\tilde{\mathcal{O}}(\min\{\sqrt{QT}, \Delta^{1/3}T^{2/3}\})$, where $Q$ and $\Delta$ denote the number and amount of changes in the cost function respectively. This is comparable to Theorem 2 when $Q$ grows faster than $\tilde{\mathcal{O}}(T^{1/4})$. Still, the bound with the change parameter satisfying $\Delta(t) \ge \max_{\pi \in \Pi} |c_t(\pi) - c_{t+1}(\pi)|$ is pessimistic while our results can still lead an optimistic bound. To see this, consider a predictor designed with the cost suffered in the last episode: i.e., $M_{t+1}(\pi_t) = c_t(\pi_t)$. Then, the optimistic bound becomes $\sigma_t = |M_t(\bar{\pi}) - c_t(\bar{\pi})|$, where $\bar{\pi}$ is a policy that visits all state-action pair once, and is a special case with the specific choice of the predictor.

Finally, we present our main result, which establishes a high probability sublinear optimistic regret bound for OREPS-OPIX.

**Theorem 3** (Bandit – High probability). *Under bandit feedback, there exists a stepsize $\eta$ and an exploration parameter $\gamma$ such that with probability $1 - \delta$ OREPS-OPIX utilizing the proposed cost estimator* (5) *satisfies*

$$
\begin{aligned}
\mathcal{R}_T(\rho^*, \{c_t\}_{t=1}^T) = \tilde{\mathcal{O}}\Bigg( &\sqrt{\sum_{t=1}^T \|\sigma_t\|_1^2} \quad (10) \\
&+ \left( L \max_t \|\sigma_t\|_\infty \right)^{\frac{1}{4}} \left( \sum_{t=1}^T \|\sigma_t\|_\infty^2 + \|\sigma_t\|_1 \right)^{\frac{3}{4}} \Bigg).
\end{aligned}
$$

We point out that the regret is, again, optimistic as it scales with the estimation power of the cost predictors. Therefore, in the best case, i.e., under perfect estimation, the regret is constant while in the worst case, the regret is $\tilde{\mathcal{O}}(T^{3/4})$, with high probability. Integral to establishing this result is the development of tailored technical lemmas and a new concentration inequality to ensure each of the individual terms in the regret remains optimistic. Further study to see the possibility of improving the regret to $\tilde{\mathcal{O}}(\sqrt{T})$ is left as a future work. Lee et al. [2020] studies the AMDP setting and achieves a high probability guarantee with sublinear regret in the order of $\sqrt{T}$ using the log-barrier method instead of implicit exploration. However, their bound $\mathcal{O}\left(\sqrt{\langle \rho^*, \sum_{t=1}^T c_t \rangle}\right)$ is in terms of the loss of the best policy as opposed to being optimistic while our bound $\mathcal{O}\left(d(\{c_t\}_{t=1}^T, \{M_t\}_{t=1}^T)^{3/4}\right)$ diminishes with the estimation power of cost predictors.

**Proof highlights.** Here we highlight the key steps towards establishing our main results stated in Theorem 3. The regret

can be decomposed into

$$
\begin{aligned}
\mathcal{R}_T&(\rho^*, \{c\}_{t=1}^T) \\
&= \sum_{t=1}^{T} \langle \rho_t - \rho^*, \hat{c}_t \rangle + \sum_{t=1}^{T} \langle \rho_t, c_t - \mathbb{E}_{t-1}[\hat{c}_t] \rangle \\
&\quad + \sum_{t=1}^{T} \langle \rho_t, \mathbb{E}_{t-1}[\hat{c}_t] - \hat{c}_t \rangle + \sum_{t=1}^{T} \langle \rho^*, \hat{c}_t - c_t \rangle.
\end{aligned}
\tag{11}
$$

The first term in (11) can be thought of as the regret of the proposed algorithm with full information when the sequence of the cost functions are $\{\hat{c}_t\}_{t=1}^T$. Hence we can use Theorem 1 as well as the result of Lemma 1 to upper bound it with probability one according to

$$
\sum_{t=1}^{T} \langle \rho_t - \rho^*, \hat{c}_t \rangle \le \frac{L}{\eta} \log \frac{|\mathcal{X}||\mathcal{A}|}{L} + \frac{\eta}{2\gamma^2} \sum_{t=1}^{T} \|\sigma_t\|_\infty^2.
$$

We then show that the second term can be bound with probability one using the definition of the proposed estimator (5) and the result of Lemma 1 with

$$
\sum_{t=1}^{T} \langle \rho_t, c_t - \mathbb{E}_{t-1}[\hat{c}_t] \rangle \le \sum_{t=1}^{T} \gamma \|\sigma_t\|_1.
$$

To bound the third term, we show that it is the sum of a martingale difference sequence, hence by using the Azuma–Hoeffding inequality and a careful computation we can bound it with probability at least $1-\delta$ with an optimistic term:

$$
\sum_{t=1}^{T} \langle \rho_t, \mathbb{E}_{t-1}[\hat{c}_t] - \hat{c}_t \rangle \le \sqrt{2 \log \frac{1}{\delta} \sum_{t=1}^{T} \|\sigma_t\|_1^2}.
$$

Notably, this term is independent of $\eta$ and $\gamma$ and in the worst case scales as $\mathcal{O}(\sqrt{T})$.

The last term in (11) requires the development of a new Bernstein-type inequality (See Lemma 2 in the supplementary) to ensure this term can be bounded by an optimistic term. Using this new result we show that with probability at least $1-\delta$

$$
\sum_{t=1}^{T} \langle \rho^*, \hat{c}_t - c_t \rangle \le \frac{L}{\gamma} \log \frac{L}{\delta} \max_{t=1,\dots,T} \|\sigma_t\|_\infty.
$$

Finally, optimizing for $\eta$ and setting $\gamma = \eta^{1/3}$ furnishes the proof of Theorem 3.

## 5 EXTENSION

### 5.1 ANYTIME OPTIMISTIC REGRET BOUNDS

In this section, we discuss the extension of OREPS-OPIX to the anytime setting. To obtain the regret bounds in Section 4,

---

**Algorithm 2** Anytime OREPS-OPIX with Doubling Trick

---

**Require:** Initial learning rate $\eta_0$, $\kappa = 2$ (expected regret) or $\kappa = 3$ (high probability regret)

1: Initialize phase number $i = 1$, starting episode number $s_1 = 1$, learning rate $\eta_1 = \eta_0/2$ and optimistic parameter $\gamma_1 = \eta_1^{1/\kappa}$
2: **for** Episodes $t = 1, 2, \dots$ **do**
3: Interact with the environment and suffer the cost to compute $\Psi_{s_i:t}$
4: **if** $\eta_i^{-1} D_0 < \eta_i^{1/\kappa} \Psi_{s_i:t}$ **then**
5:  $i \leftarrow i + 1$
6:  $s_i \leftarrow t$
7:  $\eta_i \leftarrow 2^{-i} \eta_0$
8:  $\gamma_i \leftarrow \eta_i^{1/\kappa}$
9: **end if**
10: Run the rest of Algorithm 1 to compute $\hat{c}_t$, $M_{t+1}$ and $\rho_{t+1}$ using $\eta_i$ and $\gamma_i$
11: **end for**

---

we have to utilize stepsize and exploration parameters that require the knowledge of typically unknown quantities, e.g., the horizon $T$. We alleviate this issue by utilizing the doubling trick technique [Besson and Kaufmann, 2018]. Note that compared to typical applications of the doubling trick, our setting necessitates further efforts. In particular, usually in the doubling trick the learning is divided into phases that double in length, and accordingly the stepsize is divided in half to compensate for the growing phase lengths. That is, the condition to decide when a particular phase ends is apparent. In our setting, similar to Rakhlin and Sridharan [2013], this condition is more involved as we outline next. Additionally, compared to Rakhlin and Sridharan [2013], given the more complicated setting of our problem and the intricate nature of the regret bounds, carrying out the doubling trick technique requires further innovations, especially for the high probability results. As discussed, similar to the standard doubling trick [Besson and Kaufmann, 2018, Lattimore and Szepesvári, 2018, Rakhlin and Sridharan, 2013], the learning rate $\eta_i$ is reduced by half after every phase $i$ instead of a fixed $\eta$ that depends on $T$. However, the length of each phase does not necessarily double.

Let us first consider the setting of Theorem 2. Let $D_0 = L \log \frac{|\mathcal{X}||\mathcal{A}|}{L}$ and $\bar{c}_t(x, a)$ be an unbiased cost estimator, i.e., Equation (5) with $\gamma = 0$. And define $\Psi_{\tau:\tau'} = \sum_{t=\tau}^{\tau'} \left\{ \|\bar{c}_t - M_t\|_2^2/2 + \|\bar{c}_t - M_t\|_1 \right\}$. Note that $\mathbb{E}[\Psi_{1:T}] = \sum_{t=1}^{T} \frac{1}{2}\|c_t - M_t\|_2^2 + \|c_t - M_t\|_1$.

The reason to define $\Psi$ in this way is to use it (in addition to $D_0$) to determine when to terminate each phase (see step 4 in Algorithm 2). Therefore, $\Psi$ must only contain information that is available to the learner. Since the optimistic regret bounds, naturally, depend on $c_t$ which is unknown in the bandit setting, directly utilizing the optimistic regret bound from Theorem 2 is not feasible. This subtle reason as

well as the different feedback model of our setting results in *significantly* different anytime algorithms and analyses compared to Rakhlin and Sridharan [2013].

The above discussion leads to an anytime extension of OREPS-OPIX which is summarized in Algorithm 2. This method satisfies the following expected regret bound under bandit feedback, which is comparable to Theorem 2.

**Theorem 4** (Anytime – Bandit – Expected). *Under bandit feedback, there exists an initial stepsize $\eta_0$ such that Algorithm 2 with the exploration parameter $\gamma_i = \sqrt{\eta_i}$ satisfies*

$$
\begin{aligned}
&\mathbb{E}[\mathcal{R}_T(\rho^*, \{c_t\}_{t=1}^T)] \\
&= \tilde{\mathcal{O}}\left( L^{1/3} \left( \sum_{t=1}^T \frac{\|\sigma_t\|_2^2}{2} + \|\sigma_t\|_1 \right)^{2/3} \right).
\end{aligned} \tag{12}
$$

In the full information setting, a similar doubling trick can be applied by comparing $\eta_i^{-1} D_0$ and $\eta_i \Psi_{s_i:t}$, where $\Psi_{\tau:\tau'} = \sum_{t=\tau}^{\tau'} \|c_t - M_t\|_\infty^2 / 2$. Since here $c_t$ is observed by the learner, we can directly leverage the bound from Theorem 1

**Theorem 5** (Anytime – Full information). *Under full information feedback, there exists an initial stepsize $\eta_0$ such that Algorithm 2 satisfies*

$$
\mathcal{R}_T(\rho^*, \{c_t\}_{t=1}^T) = \tilde{\mathcal{O}}\left( \sqrt{L \sum_{t=1}^T \|\sigma_t\|_\infty^2} \right). \tag{13}
$$

## 5.2 HANDLING UNKNOWN TRANSITION

In this section, we extend our prior results to the unknown transition setting. This allows the algorithm the flexibility to be used when the dynamics of MDP is not revealed to the learner. To model the unknown transition, we construct a confidence set of transition functions using the counting method as explored by Jaksch et al. [2010], Azar et al. [2017], Rosenberg and Mansour [2019], Jin et al. [2020]. Specifically, we adopt a tighter confidence set from Jin et al. [2020, Equation 5]:

$$
\begin{aligned}
\mathcal{P} = \Big\{ \hat{P} : &\left| \hat{P}(x'|x,a) - \bar{P}(x'|x,a) \right| \le \epsilon(x'|x,a), \\
&\forall (x,a,x') \in \mathcal{X}_k \times \mathcal{A} \times \mathcal{X}_{k+1}, k \in (0, L-1) \Big\}
\end{aligned} \tag{14}
$$

where $\bar{P}$ is the count-based empirical transition probability and the confidence margin $\epsilon(x'|x,a)$ is defined as

$$
2\sqrt{\frac{\bar{P}(x'|x,a)\log\left(\frac{T|\mathcal{X}||\mathcal{A}|}{\delta}\right)}{\max\{1, N(x,a)-1\}}} + \frac{14\log\left(\frac{T|\mathcal{X}||\mathcal{A}|}{\delta}\right)}{3\max\{1, N(x,a)-1\}}
$$

for $\delta \in (0,1)$ and state-action visit counter $N(x,a)$. And we propose a cost estimator as

$$
\begin{aligned}
&\hat{c}_t(x,a) \\
&= \frac{c_t(x,a) - M_t(x,a)}{u_t(x,a) + \gamma} \mathbb{I}\{(x,a) \in \bar{\mathbf{u}}_L(t)\} + M_t(x,a),
\end{aligned} \tag{15}
$$

where $u_t(x,a) = \max_{P \in \mathcal{P}} \rho^{P,\pi_t}(x,a)$ is the upper occupancy bound over $\mathcal{P}$ and $\rho^{P,\pi}$ is the occupancy measure under the transition probability $P$ and the induced policy $\pi_t$ from $\rho_t$ as (1). Again, (15) is an optimistically biased estimator given that the predictor is optimistic and $u_t(x,a) \ge \rho_t(x,a)$ by definition. Utilizing this new estimator in OREPS-OPIX, we obtain the following result.

**Theorem 6** (Unknown transition – Bandit – High probability). *Under bandit feedback with unknown transition, there exists a stepsize $\eta$ and an exploration parameter $\gamma$ such that with probability at least $1 - 7\delta$ OREPS-OPIX utilizing the proposed cost estimator* (15) *satisfies*

$$
\begin{aligned}
&\mathcal{R}_T(\rho^*, \{c_t\}_{t=1}^T) \\
&= \mathcal{O}\Bigg( L^{\frac{1}{4}} \left( \log\frac{|\mathcal{X}||\mathcal{A}|}{L} + \log\frac{L}{\delta} \max_t \|\sigma_t\|_\infty \right)^{\frac{1}{4}} \\
&\quad \cdot \left( \sum_{t=1}^T \|\sigma_t\|_\infty^2 + \|\sigma_t\|_1 \right)^{\frac{3}{4}} + \sqrt{\sum_{t=1}^T \|\sigma_t\|_1^2} \\
&\quad + L|\mathcal{X}|\sqrt{|\mathcal{A}|T\log\frac{T|\mathcal{X}||\mathcal{A}|}{\delta}} \Bigg).
\end{aligned} \tag{16}
$$

Notice that in an optimistic case, the bound is dominated by the term $\mathcal{O}\left( L|\mathcal{X}|\sqrt{|\mathcal{A}|T\log\frac{T|\mathcal{X}||\mathcal{A}|}{\delta}} \right)$. Then the Theorem 6 achieves the same bound as Jin et al. [2020] but with higher probability. This term arises from a judicious application of the Bennet's concentration inequality [Maurer and Pontil, 2009, Corollary 5] to study how the error of the estimated occupancy measure $\rho^{P,\pi_t}$ with respect to $\rho_t$ of known transition setting is bounded within the confidence set (14); it is nontrivial and therefore an interesting direction of research to see if an optimistic version of this concentration inequality can be established, using, e.g., the techniques that led to our new Bernstein-type inequality (See Lemma 2 in Appendix A.4).

## 6 NUMERICAL EXPERIMENTS

In this section, we perform a simple experiment to demonstrate the benefit of implicit exploration and cost predictors. [3] We consider a drone navigation task modeled by a 2D grid, where the goal of the agent is to move by one cell at a time

---

[3]The code for this experiment is accessible at this link: https://github.itap.purdue.edu/moon182/OREPS-OPIX.git

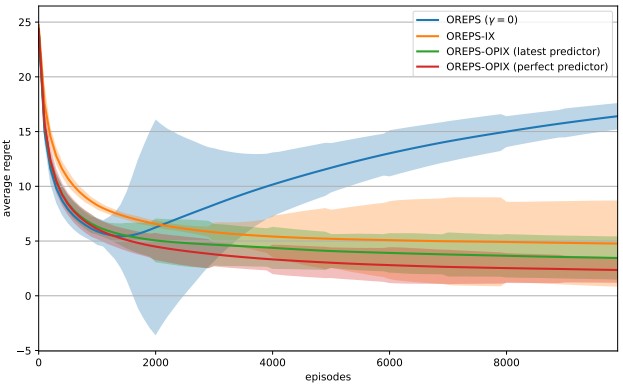

(a) Average regret and variance of OREPS-OPIX, OREPS, and OREPS-IX.

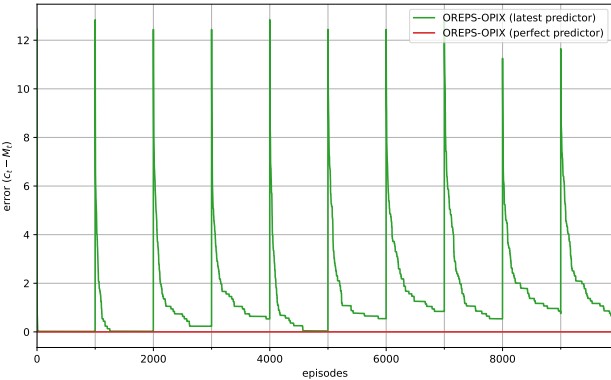

(b) Error of cost predictors against the true cost function.

Fig. 1: The result of numerical experiment of OREPS, OREPS-IX and OREPS-OPIX with different predictors plotted versus the number of episodes. Figure 1(a) shows the regret reduction benefit as well as the variance reduction property of the proposed cost estimator (5). Figure 1(b) shows that the cost predictors comply with the optimistic prediction assumption.

to reach the goal with minimal cost. If a drone enters a cell with turbulence or wind gust, it incurs higher cost due to higher fuel consumption and possible damage to the aircraft. The AMDP of the environment is described below:

- State space: $\mathcal{X} = \{(l, A_x, A_y, G_x, G_y)\}$, where $l \in \{1, \ldots, L\}$ is time step, $(A_x, A_y)$ is agent location and $(G_x, G_y)$ is goal location.
- Action space: $\mathcal{A} = \{\text{left, right, up, down}\}$
- Cost function:
$$c_t(x, a) = \begin{cases} 0, & \text{if reaching the goal} \\ 1, & \text{if encountering a turbulence} \\ \epsilon, & \text{otherwise}, \end{cases}$$
where $0 < \epsilon < 1$ is a small positive constant. $c_t$ changes every $t_w$ episodes when the occurrence of turbulence randomly move to one of its neighbors. It is not observable to the agent but results in higher cost.
- Bandit feedback: agent observes $c_t(x, a)$ only for its trajectory $(x, a) \in \mathbf{u}(t)$ in episode $t$.
- State transition is deterministic:
$$\Pr(s'|s, a) = \begin{cases} 1, & \text{when (x,a) results in s'} \\ 0, & \text{otherwise.} \end{cases}$$
- Wind incurs cost but does not affect state transitions.
- Timeout $L$ is the maximum time steps in an episode.
- When the agent reaches the goal, it remains in that terminal state $s^l_{\text{terminal}}$ until the end of the episode regardless of its action, that is, $\mathbb{P}(s^{l+1}_{\text{terminal}}|s^l_{\text{terminal}}, \mathcal{A}) = 1$ and $\mathcal{X}_L = \{s^L_{\text{terminal}}\}$ is singleton.

The details of the experiment setting are provided in the Appendix.

Figure 1(a) depicts the performance (in terms of cumulative average regret) of OREPS-OPIX compared with vanilla

OREPS and OREPS with implicit exploration. For OREPS-OPIX with perfect predictor, it is assumed that we have access to a perfect predictor with full information ($M_t = c_t$, $M_{t+1} = c_{t+1}$). A more realistic latest predictor predicts the cost based on the cost that the learner suffered in the last visit to the state and the action. It mildly assumes that we have access to the period $t_w$ and it resets its value to zero every $t_w$ episodes to assure optimistic prediction.

There are two notable points to this result. First, OREPS without implicit exploration (in blue) explodes as learning progresses. This happens when the value of occupancy measure for some states and actions approach 0: $\rho_t(x, a) \to 0$. Then, the unbiased cost estimator, i.e., (4) with $\gamma = 0$, which divides cost signal by occupancy measure, grows infinitely large and $\rho_t(x, a)$ actually becomes 0 due to the precision of the floating point. And it remains to be 0 for the remainder of the episode, because the occupancy measure is updated multiplicatively according to (7). This phenomenon is consistent with the fact that the naive importance-weighted cost estimator in OREPS which is based on EXP3 suffers from a high variance. Secondly, OREPS-OPIX (in green and red) improves both convergence and variance over OREPS-IX (in orange), which is consistent with the result of Lemma 1 on the reduced variance of the proposed cost estimator (5) while retaining the same bias.

Figure 1(b) demonstrates the error of optimistic cost predictors with respect to the true cost. By observing the positive values of error, we confirm that the formulation of cost predictors does not violate the optimistic prediction assumption. Every $t_w = 1000$ episodes, the error of the latest predictor spikes, because it periodically resets its value to zero.

In Figure 2(a), we relax the optimistic prediction assumption with inaccurate information about how frequently the cost function changes. The latest predictor with more reset

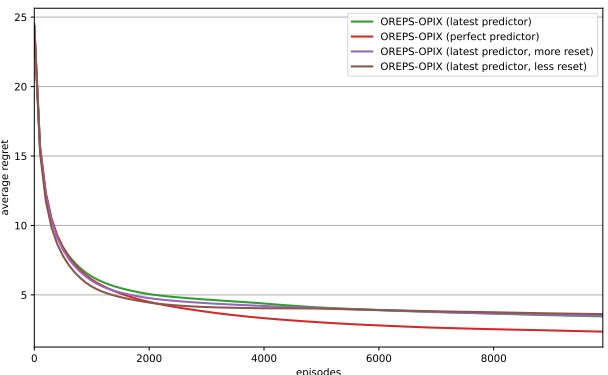
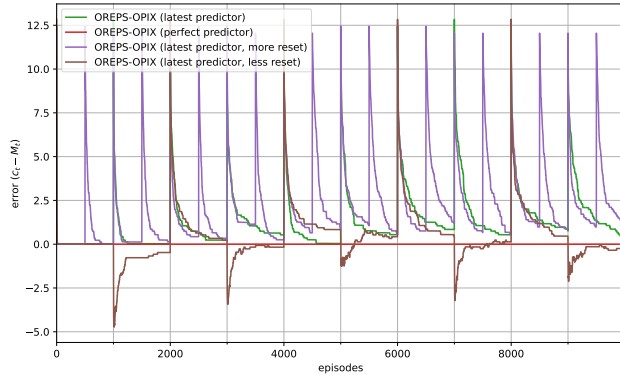

(a) Average regret of OREPS-OPIX with different predictors.

(b) Error of cost predictors against the true cost function.

Fig. 2: The result of numerical experiment of OREPS-OPIX with different predictors plotted versus the number of episodes. Figure 2(a) shows that less accurate information about $t_w$ do not cause significant harm in the performance of OREPS-OPIX. Figure 2(b) shows the consequences on error when cost predictors are constructed based on inaccurate information about the environment.

(in purple) and less reset (in brown) assumes shorter and longer periods of change, respectively, than the true value of $t_w$. However, the results show that the performance degradation is not noticeable compared to the latest predictor with accurate information about the period (in green). In fact, the strict optimism of the predictor is introduced for mathematical convenience and it is sufficient to hold in a (weighted) sum: $\sum_{x,a} \omega(x,a)(c_t(x,a) - M_t(x,a)) \geq 0$ with $\omega(\cdot) = 1$ or $\omega(\cdot) > 0$. Intuitively, what is more critical is how far the prediction is to the true cost function.

Figure 2(b) shows the error, i.e., $\sum_{x,a} c_t(x,a) - M_t(x,a)$, of different predictors. The latest predictor with more reset and less reset is built based on incorrect information of the period of cost change, as $\hat{t}_m = 500$ and $\hat{t}_m = 2000$ respectively. Although Figure 2(a) demonstrates minimal loss in the performance of OREPS-OPIX when predictor design is based on a flawed information, Figure 2(b) shows that the predictor error is actually aggravated by the flaws (purple and brown as opposed to green). It even shows that the latest predictor with less reset (brown) violates the optimistic prediction assumption when cost function changes without the reset, observed at $t = 1000, 3000, \ldots, 9000$. The result hints at the practical success of our algorithm in the presence of minor uncertainties in the predictor design.

Finally, Figures 1(b) and 2(b) also exhibits a tendency that the error grows higher over time as the occupancy measure converges. It is the result of slower convergence of $M_t$, which is caused by the reduced entropy of the occupancy measure. Equation (7) updates the occupancy measure by discounting its value exponentially with respect to the loss (estimate) and forces the value of a state-action pair with relatively high loss (estimate) to approach to zero. From the OREPS regret plot (blue) in Figure 1(a), the exploding regret is also observed, that is due to the fact that a state-

action pair with near-zero occupancy measure cannot be visited again without implicit exploration.

## 7 CONCLUSION

We studied the problem of establishing optimistic regret bounds for online learning in AMDPs. Our theoretical analysis demonstrated that such bounds in the bandit feedback setting necessitate cost estimators with a bounded variance that scales with the estimation power of cost predictors. To that end, we proposed a new estimator that benefits from variance reduction and proved that this estimator in conjunction with a variant of mirror descent enjoys optimistic regret bounds in both full information and bandit feedback settings. Notably, we showed the proposed method and its anytime extension enjoy high probability sublinear optimistic regrets, a result which crucially relied on the characteristics of the new cost estimator and the development of new technical lemmas to ensure every term in the regret decomposition can be bounded by optimistic terms. Finally, we provided an extension to the unknown transition setting and established similar results.

In MDP setting, the cost function remains constant over time and direct optimization of the cost function without bounding the relative entropy becomes feasible. In the case of full information feedback, the cost function is fully observed after the initial episode, resulting in zero regret from the second episode onward. In bandit feedback case, we have a bound with diminishing prediction error $c_t - M_t$, as costs are revealed for additional states and actions. The rate at which the error reduces and efficient strategies for its reduction present an interesting direction for future research.

**Acknowledgements**

This work was supported in part by NSF CNS 2313109 and the Manufacturing Design Laboratory (MDLab) at Purdue University.

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

# Optimistic Regret Bounds for Online Learning in Adversarial Markov Decision Processes (Supplementary Material)

**Sang Bin Moon**[1]                    **Abolfazl Hashemi**[1]

[1]School of Electrical and Computer Engineering, Purdue University, West Lafayette, Indiana, USA

## A  PROOFS

### A.1  PROOF OF LEMMA 1

By the definition of $\hat{c}(x,a)$ as (5),

$$
\begin{aligned}
\mathbb{E}_{t-1}[\hat{\boldsymbol{c}}_t(x,a)] &= \mathbb{E}_{t-1}\left[\frac{c_t(x,a) - M_t(x,a)}{\rho_t(x,a) + \gamma}\mathbb{I}\{(x,a) \in \bar{\mathbf{u}}_L(t)\} + M_t(s,a)\right] \\
&= \frac{c_t(x,a) - M_t(x,a)}{\rho_t(x,a) + \gamma}\rho_t(x,a) + M_t(s,a) \\
&= \frac{\rho_t(x,a)c_t(x,a) + \gamma M_t(x,a)}{\rho_t(x,a) + \gamma}.
\end{aligned}
$$

By (5), $\rho_t(\cdot) \geq 0$ and $\gamma \geq 0$,

$$
\begin{aligned}
\mathbb{E}_{t-1}[(\hat{\boldsymbol{c}}_t(x,a) - M_t(x,a))^2] &= \mathbb{E}_{t-1}\left[\left(\frac{c_t(x,a) - M_t(x,a)}{\rho_t(x,a) + \gamma}\mathbb{I}\{(x,a) \in \bar{\mathbf{u}}_L(t)\}\right)^2\right] \\
&= \left(\frac{c_t(x,a) - M_t(x,a)}{\rho_t(x,a) + \gamma}\right)^2\rho_t(x,a) \\
&\leq \frac{(c_t(x,a) - M_t(x,a))^2}{\rho_t(x,a) + \gamma}.
\end{aligned}
$$

∎

### A.2  PROOF OF THEOREM 1

First, decompose the regret of $\rho_t$ with respect to $\rho^*$ as

$$
\langle c_t, \rho_t - \rho^* \rangle = \langle c_t, \rho_{t+1} - \rho^* \rangle + \langle c_t, \rho_t - \rho_{t+1} \rangle. \tag{17}
$$

If $\rho_{t+1}$ is the solution of (6) with $c_t$ instead of $\hat{c}$, then for any other $\rho^* \in \Delta(\mathcal{M})$, the gradient of the objective function is negative in the direction of $\rho_{t+1}$ from $\rho^*$: i.e., $\langle \nabla_\rho\{\eta\langle\rho, c_t + M_{t+1} - M_t\rangle + D_R(\rho\|\rho_t)\}_{\rho=\rho_{t+1}}, \rho_{t+1} - \rho^* \rangle \leq 0$. Thus,

$$
\langle \eta(c_t + M_{t+1} - M_t) + \nabla R(\rho_{t+1}) - \nabla R(\rho_t), \rho_{t+1} - \rho^* \rangle \leq 0.
$$

The first term of the decomposition (17) is then bounded as

$$
\langle c_t, \rho_{t+1} - \rho^* \rangle \leq \frac{1}{\eta}\langle \nabla R(\rho_t) - \nabla R(\rho_{t+1}), \rho_{t+1} - \rho^* \rangle + \langle M_t - M_{t+1}, \rho_{t+1} - \rho^* \rangle.
$$

By the definition of Bregman divergence: $D_R(\rho\|\rho') = R(\rho) - R(\rho') - \langle\nabla R(\rho'), \rho - \rho'\rangle$,

$$\langle c_t, \rho_{t+1} - \rho^*\rangle \leq \frac{1}{\eta}\{D_R(\rho^*\|\rho_t) - D_R(\rho^*\|\rho_{t+1}) - D_R(\rho_{t+1}\|\rho_t)\} + \langle M_t - M_{t+1}, \rho_{t+1} - \rho^*\rangle$$

$$= \frac{1}{\eta}\{D_R(\rho^*\|\rho_t) - D_R(\rho^*\|\rho_{t+1}) - D_R(\rho_{t+1}\|\rho_t)\} + \langle M_{t+1} - M_t, \rho^* - \rho_{t+1}\rangle$$

$$= \frac{1}{\eta}\{D_R(\rho^*\|\rho_t) - D_R(\rho^*\|\rho_{t+1}) - D_R(\rho_{t+1}\|\rho_t)\}$$
$$+ \langle M_{t+1} - M_t, \rho^* - \rho_t\rangle + \langle M_{t+1} - M_t, \rho_t - \rho_{t+1}\rangle.$$

Plugging the result back to (17),

$$\langle c_t, \rho_t - \rho^*\rangle \leq \frac{1}{\eta}\{D_R(\rho^*\|\rho_t) - D_R(\rho^*\|\rho_{t+1}) - D_R(\rho_{t+1}\|\rho_t)\}$$
$$+ \langle M_{t+1} - M_t, \rho^* - \rho_t\rangle + \langle M_{t+1}, \rho_t - \rho_{t+1}\rangle + \langle c_t - M_t, \rho_t - \rho_{t+1}\rangle$$
$$= \frac{1}{\eta}\{D(\rho^*\|\rho_t) - D(\rho^*\|\rho_{t+1}) - D(\rho_{t+1}\|\rho_t)\} \qquad (18)$$
$$- \langle M_t, \rho^* - \rho_t\rangle + \langle M_{t+1}, \rho^* - \rho_{t+1}\rangle + \langle c_t - M_t, \rho_t - \rho_{t+1}\rangle.$$

By Holder's and Young's inequalities,

$$\langle c_t - M_t, \rho_t - \rho_{t+1}\rangle \leq \frac{\eta}{2}\|c_t - M_t\|_\infty^2 + \frac{1}{2\eta}\|\rho_t - \rho_{t+1}\|_1^2.$$

Since negative entropy is 1-strongly convex with respect to $L_1$ norm,

$$\frac{1}{2}\|\rho_t - \rho_{t+1}\|_1^2 \leq R(\rho_{t+1}) - R(\rho_t) - \langle\nabla R(\rho_t), \rho_{t+1} - \rho_t\rangle = D_R(\rho_{t+1}\|\rho_t).$$

Plugging the result back to (18),

$$\langle c_t, \rho_t - \rho^*\rangle \leq \frac{1}{\eta}\{D(\rho^*\|\rho_t) - D(\rho^*\|\rho_{t+1}) - D(\rho_{t+1}\|\rho_t)\}$$
$$- \langle M_t, \rho^* - \rho_t\rangle + \langle M_{t+1}, \rho^* - \rho_{t+1}\rangle + \frac{\eta}{2}\|c_t - M_t\|_\infty^2 + \frac{1}{\eta}D(\rho_{t+1}\|\rho_t)$$
$$= \frac{1}{\eta}\{D(\rho^*\|\rho_t) - D(\rho^*\|\rho_{t+1})\} - \langle M_t, \rho^* - \rho_t\rangle + \langle M_{t+1}, \rho^* - \rho_{t+1}\rangle$$
$$+ \frac{\eta}{2}\|c_t - M_t\|_\infty^2.$$

By summing over $T$ episodes,

$$\mathcal{R}_T(\rho^*, \{c_t\}_{t=1}^T) = \sum_{t=1}^T \langle c_t, \rho_t - \rho^*\rangle$$
$$\leq \frac{1}{\eta}\{D(\rho^*\|\rho_1) - D(\rho^*\|\rho_{T+1})\} - \langle M_1, \rho^* - \rho_1\rangle + \langle M_{T+1}, \rho^* - \rho_{T+1}\rangle$$
$$+ \sum_{t=1}^T \frac{\eta}{2}\|c_t - M_t\|_\infty^2.$$

Without losing generality, we can set $M_1 = M_{T+1} = 0$. And by the non-negativity and definition of Bregman divergence,

$$\mathcal{R}_T(\rho^*, \{c_t\}_{t=1}^T) \leq \frac{1}{\eta}D(\rho^*\|\rho_1) + \sum_{t=1}^T \frac{\eta}{2}\|c_t - M_t\|_\infty^2$$
$$= \frac{1}{\eta}\{R(\rho^*) - R(\rho_1) - \langle\nabla R(\rho_1), \rho^* - \rho_1\rangle\} + \sum_{t=1}^T \frac{\eta}{2}\|c_t - M_t\|_\infty^2.$$

Since negative entropy $R(\cdot) \leq 0$ and $\rho_1$ is initialized as a uniform distribution, of which $\nabla R(\rho_1) = 0$,

$$\mathcal{R}_T(\rho^*, \{c_t\}_{t=1}^T) \leq -\frac{1}{\eta} R(\rho_1) + \sum_{t=1}^{T} \frac{\eta}{2} \|c_t - M_t\|_\infty^2$$

$$= \frac{1}{\eta} \sum_{k=0}^{L-1} \sum_{x \in \mathcal{X}_k} \sum_a (-\rho_1(x,a) \log(\rho_1(x,a))) + \sum_{t=1}^{T} \frac{\eta}{2} \|c_t - M_t\|_\infty^2$$

$$= \frac{1}{\eta} \sum_{k=0}^{L-1} \sum_{x \in \mathcal{X}_k} \sum_a \frac{1}{|\mathcal{X}_k||\mathcal{A}|} \log |\mathcal{X}_k||\mathcal{A}| + \sum_{t=1}^{T} \frac{\eta}{2} \|c_t - M_t\|_\infty^2$$

$$= \frac{1}{\eta} \sum_{k=0}^{L-1} \log |\mathcal{X}_k||\mathcal{A}| + \sum_{t=1}^{T} \frac{\eta}{2} \|c_t - M_t\|_\infty^2$$

$$= \frac{L}{\eta} \log \frac{|\mathcal{X}||\mathcal{A}|}{L} + \sum_{t=1}^{T} \frac{\eta}{2} \|c_t - M_t\|_\infty^2.$$

If $\eta = \sqrt{\frac{2L}{\sum \|c_t - M_t\|_\infty^2} \log \frac{|\mathcal{X}||\mathcal{A}|}{L}}$,

$$\mathcal{R}_T(\rho^*, \{c_t\}_{t=1}^T) \leq \sqrt{2L \log \frac{|\mathcal{X}||\mathcal{A}|}{L} \sum_{t=1}^{T} \|c_t - M_t\|_\infty^2}.$$

∎

## A.3 PROOF OF THEOREM 2

The expected total regret with respect to $\rho^* \in \Delta(\mathcal{M})$ can be decomposed into

$$\mathbb{E}\left[\sum_{t=1}^{T} \langle \rho_t - \rho^*, c_t \rangle \right] = \mathbb{E}\left[\sum_{t=1}^{T} \langle \rho_t - \rho^*, \hat{c}_t \rangle \right] + \mathbb{E}\left[\sum_{t=1}^{T} \langle \rho_t, c_t - \hat{c}_t \rangle \right] - \mathbb{E}\left[\sum_{t=1}^{T} \langle \rho^*, c_t - \hat{c}_t \rangle \right]. \tag{19}$$

For the first term in (19), follow the same proof as Appendix A.2 with $c_t \leftarrow \hat{c}_t$. Let $\rho_{t+1}$ be the solution of (6) and decompose the term as

$$\langle \rho_t - \rho^*, \hat{c}_t \rangle = \langle \rho_{t+1} - \rho^*, \hat{c}_t \rangle + \langle \rho_t - \rho_{t+1}, \hat{c}_t \rangle.$$

For all $\rho^* \in \Delta(\mathcal{M})$, the gradient of (6) is negative in the direction of $\rho_{t+1}$ from $\rho^*$: i.e., $\langle \nabla_\rho \{ \eta \langle \rho, \hat{c}_t + M_{t+1} - M_t \rangle + D(\rho\|\rho_t)\}_{\rho=\tilde{\rho}_{t+1}}, \rho_{t+1} - \rho^* \rangle \leq 0$. Thus, following Appendix A.2,

$$\langle \rho_t - \rho^*, \hat{c}_t \rangle \leq \frac{1}{\eta} \langle \rho_{t+1} - \rho^*, \nabla R(\rho_t) - \nabla R(\rho_{t+1}) \rangle + \langle \rho_{t+1} - \rho^*, M_t - M_{t+1} \rangle + \langle \rho_t - \rho_{t+1}, \hat{c}_t \rangle$$

$$\leq \frac{1}{\eta} \{ D(\rho^*\|\rho_t) - D(\rho^*\|\rho_{t+1}) \} - \langle M_t, \rho^* - \rho_t \rangle + \langle M_{t+1}, \rho^* - \rho_{t+1} \rangle + \frac{\eta}{2} \|\hat{c}_t - M_t\|_\infty^2.$$

Adding over $t$ episodes, following Appendix A.2 again,

$$\sum_{t=1}^{T} \langle \rho_t - \rho^*, \hat{c}_t \rangle \leq \frac{L}{\eta} \log \frac{|\mathcal{X}||\mathcal{A}|}{L} + \sum_{t=1}^{T} \frac{\eta}{2} \|\hat{c}_t - M_t\|_\infty^2.$$

Taking expectation over the randomness associated with $\mathbf{u}(T)$,

$$\mathbb{E}\left[\sum_{t=1}^{T} \langle \rho_t - \rho^*, \hat{c}_t \rangle \right] \leq \mathbb{E}\left[\frac{L}{\eta} \log \frac{|\mathcal{X}||\mathcal{A}|}{L} + \sum_{t=1}^{T} \frac{\eta}{2} \|\hat{c}_t - M_t\|_\infty^2 \right]$$

$$= \frac{L}{\eta} \log \frac{|\mathcal{X}||\mathcal{A}|}{L} + \frac{\eta}{2} \sum_{t=1}^{T} \mathbb{E}\left[\|\hat{c}_t - M_t\|_\infty^2 \right].$$

By tower expectation,

$$\mathbb{E}\left[\sum_{t=1}^{T}\langle\rho_t - \rho^*, \hat{c}_t\rangle\right] \leq \frac{L}{\eta}\log\frac{|\mathcal{X}||\mathcal{A}|}{L} + \frac{\eta}{2}\sum_{t=1}^{T}\mathbb{E}\left[\mathbb{E}_{t-1}[\|\hat{c}_t - M_t\|_\infty^2]\right].$$

By $\|\cdot\|_\infty^2 \leq \|\cdot\|_2^2$ and Lemma 1,

$$\mathbb{E}\left[\sum_{t=1}^{T}\langle\rho_t - \rho^*, \hat{c}_t\rangle\right] \leq \frac{L}{\eta}\log\frac{|\mathcal{X}||\mathcal{A}|}{L} + \frac{\eta}{2}\sum_{t=1}^{T}\mathbb{E}\left[\mathbb{E}_{t-1}[\|\hat{c}_t - M_t\|_2^2]\right]$$

$$= \frac{L}{\eta}\log\frac{|\mathcal{X}||\mathcal{A}|}{L} + \frac{\eta}{2}\sum_{t=1}^{T}\mathbb{E}\left[\sum_{x,a}\mathbb{E}_{t-1}[(\hat{c}_t(x,a) - M_t(x,a))^2]\right]$$

$$\leq \frac{L}{\eta}\log\frac{|\mathcal{X}||\mathcal{A}|}{L} + \frac{\eta}{2}\sum_{t=1}^{T}\mathbb{E}\left[\sum_{x,a}\frac{(c_t(x,a) - M_t(x,a))^2}{\rho_t(x,a) + \gamma}\right].$$

Since $\rho_t(\cdot) \geq 0$ and $\gamma > 0$,

$$\mathbb{E}\left[\sum_{t=1}^{T}\langle\rho_t - \rho^*, \hat{c}_t\rangle\right] \leq \frac{L}{\eta}\log\frac{|\mathcal{X}||\mathcal{A}|}{L} + \frac{\eta}{2\gamma}\sum_{t=1}^{T}\|c_t - M_t\|_2^2. \qquad (20)$$

The second term in (19) can be decomposed into

$$\langle\rho_t, c_t - \hat{c}_t\rangle = \langle\rho_t, c_t - \mathbb{E}_{t-1}[\hat{c}_t]\rangle + \langle\rho_t, \mathbb{E}_{t-1}[\hat{c}_t] - \hat{c}_t\rangle.$$

By Lemma 1,

$$\langle\rho_t, c_t - \mathbb{E}_{t-1}[\hat{c}_t]\rangle = \sum_{x,a}\rho_t(x,a)\left\{c_t(x,a) - \frac{\rho_t(x,a)c_t(x,a) + \gamma M_t(x,a)}{\rho_t(x,a) + \gamma}\right\}$$

$$= \sum_{x,a}\rho_t(x,a)\frac{\gamma c_t(x,a) - \gamma M_t(x,a)}{\rho_t(x,a) + \gamma}.$$

Since $\rho_t(\cdot) \geq 0$, $\gamma > 0$ and $M_t(x,a) \leq c_t(x,a)$ for all $x, a$,

$$\langle\rho_t, c_t - \mathbb{E}_{t-1}[\hat{c}_t]\rangle \leq \sum_{x,a}\gamma|c_t(x,a) - M_t(x,a)| = \gamma\|c_t - M_t\|_1.$$

Adding over $T$ episodes and taking expectation with respect to the randomness associated with $\mathbf{u}(T)$,

$$\mathbb{E}\left[\sum_{t=1}^{T}\langle\rho_t, c_t - \mathbb{E}_{t-1}[\hat{c}_t]\rangle\right] \leq \sum_{t=1}^{T}\gamma\|c_t - M_t\|_1.$$

Also, since $\{\mathbb{E}_{t-1}[\hat{c}_t] - \hat{c}_t\}_t$ is a Martingale difference sequence (MDS),

$$\mathbb{E}\left[\sum_{t=1}^{T}\langle\rho_t, \mathbb{E}_{t-1}[\hat{c}_t] - \hat{c}_t\rangle\right] = 0.$$

Thus, the second term in (19) is bounded as

$$\mathbb{E}\left[\sum_{t=1}^{T}\langle\rho_t, c_t - \hat{c}_t\rangle\right] \leq \gamma\sum_{t=1}^{T}\|c_t - M_t\|_1. \qquad (21)$$

Finally, since $\rho^*$ is constant with respect to $t$ and the randomness associated with $\mathbf{u}(T)$,

$$\mathbb{E}\left[\sum_{t=1}^{T}\langle\rho^*, c_t - \hat{c}_t\rangle\right] = \left\langle\rho^*, \mathbb{E}\left[\sum_{t=1}^{T}c_t - \hat{c}_t\right]\right\rangle.$$

By tower expectation and Lemma 1,

$$\mathbb{E}\left[\sum_{t=1}^{T}\langle\rho^*, c_t - \hat{c}_t\rangle\right] = \langle\rho^*, \mathbb{E}\left[\sum_{t=1}^{T} c_t - \mathbb{E}_{t-1}[\hat{c}_t]\right]\rangle$$

$$= \sum_{x,a}\rho^*(x,a)\mathbb{E}\left[\sum_{t=1}^{T} c_t(x,a) - \frac{\rho_t(x,a)c_t(x,a) + \gamma M_t(x,a)}{\rho_t(x,a) + \gamma}\right]$$

$$= \sum_{x,a}\rho^*(x,a)\mathbb{E}\left[\sum_{t=1}^{T} \frac{\gamma(c_t - M_t(x,a))}{\rho_t(x,a) + \gamma}\right]$$

Since $\rho_t(\cdot), \rho^*(\cdot) \geq 0$, $\gamma > 0$ and $M_t(x,a) \leq c_t(x,a)$ for all $x,a$,

$$\mathbb{E}\left[\sum_{t=1}^{T}\langle\rho^*, c_t - \hat{c}_t\rangle\right] \geq 0 \tag{22}$$

Applying (20), (21) and Equation (22) to Equation (19):

$$\mathbb{E}\left[\sum_{t=1}^{T}\langle\rho_t - \rho^*, c_t\rangle\right] \leq \frac{L}{\eta}\log\frac{|\mathcal{X}||\mathcal{A}|}{L} + \frac{\eta}{2\gamma}\sum_{t=1}^{T}\|c_t - M_t\|_2^2 + \gamma\sum_{t=1}^{T}\|c_t - M_t\|_1$$

If $\eta = \left(\frac{L}{\sum\frac{1}{2}\|c_t - M_t\|_2^2 + \|c_t - M_t\|_1}\log\frac{|\mathcal{X}||\mathcal{A}|}{L}\right)^{2/3}$ and $\gamma = \sqrt{\eta}$,

$$\mathbb{E}[\mathcal{R}_T(\rho^*, \{c_t\}_{t=1}^{T})] \leq \left(L\log\frac{|\mathcal{X}||\mathcal{A}|}{L}\right)^{1/3}\left(\sum\frac{1}{2}\|c_t - M_t\|_2^2 + \|c_t - M_t\|_1\right)^{2/3}$$

∎

## A.4   PROOF OF THEOREM 3

The total regret can be decomposed as (11). The first term in (11) can be thought of as the regret of the the proposed algorithm with full information when the sequence of the cost functions are $\{\hat{c}_t\}_{t=1}^{T}$. By Theorem 1,

$$\sum_{t=1}^{T}\langle\rho_t - \rho^*, \hat{c}_t\rangle \leq \frac{L}{\eta}\log\frac{|\mathcal{X}||\mathcal{A}|}{L} + \frac{\eta}{2}\sum_{t=1}^{T}\|\hat{c}_t - M_t\|_\infty^2.$$

By (5),

$$\|\hat{c}_t - M_t\|_\infty^2 = \max_{x,a}\left(\frac{c_t(x,a) - M_t(x,a)}{\rho_t(x,a) + \gamma}\mathbb{I}\{(x,a) \in \bar{\mathbf{u}}_L(t)\}\right)^2$$

$$= \max_{(x,a)\in\bar{\mathbf{u}}_L(t)}\left(\frac{c_t(x,a) - M_t(x,a)}{\rho_t(x,a) + \gamma}\right)^2$$

$$\leq \max_{x,a}\left(\frac{c_t(x,a) - M_t(x,a)}{\rho_t(x,a) + \gamma}\right)^2.$$

By $\rho_t(\cdot) \geq 0$ and $\gamma \geq 0$,

$$\|\hat{c}_t - M_t\|_\infty^2 \leq \frac{\max_{x,a}(c_t(x,a) - M_t(x,a))^2}{\gamma^2}$$

$$= \frac{\|c_t - M_t\|_\infty^2}{\gamma^2}$$

Thus, the first term is bounded with probability one as

$$\sum_{t=1}^{T} \langle \rho_t - \rho^*, \hat{c}_t \rangle \leq \frac{L}{\eta} \log \frac{|\mathcal{X}||\mathcal{A}|}{L} + \frac{\eta}{2\gamma^2} \sum_{t=1}^{T} \|c_t - M_t\|_\infty^2. \tag{23}$$

Using Lemma 1, the second term is rewritten as

$$\sum_{t=1}^{T} \langle \rho_t, c_t - \mathbb{E}_{t-1}[\hat{c}_t] \rangle = \sum_{t=1}^{T} \sum_{x,a} \rho_t(x,a) \left( c_t(x,a) - \frac{\rho_t(x,a)c_t(x,a) + \gamma M_t(x,a)}{\rho_t(x,a) + \gamma} \right)$$

$$= \sum_{t=1}^{T} \sum_{x,a} \rho_t(x,a) \left( \frac{\gamma c_t(x,a) - \gamma M_t(x,a)}{\rho_t(x,a) + \gamma} \right)$$

By $M_t(x,a) \leq c_t(x,a)$, $\rho_t(\cdot) \geq 0$ and $\gamma \geq 0$, it is bounded with probability one with

$$\sum_{t=1}^{T} \langle \rho_t, c_t - \mathbb{E}_{t-1}[\hat{c}_t] \rangle \leq \sum_{t=1}^{T} \sum_{x,a} \gamma c_t(x,a) - \gamma M_t(x,a)$$

$$= \gamma \sum_{t=1}^{T} \|c_t - M_t\|_1. \tag{24}$$

By Lemma 1 and (5),

$$\mathbb{E}_{t-1}[\hat{c}_t(x,a)] - \hat{c}_t(x,a) = \frac{\rho_t(x,a)c_t(x,a) + \gamma M_t(x,a)}{\rho_t(x,a) + \gamma}$$

$$- \left( \frac{c_t(x,a) - M_t(x,a)}{\rho_t(x,a) + \gamma} \mathbb{I}\{(x,a) \in \bar{\mathbf{u}}_L(t)\} + M_t(x,a) \right)$$

$$= \frac{(\rho_t(x,a) - \mathbb{I}\{(x,a) \in \bar{\mathbf{u}}_L(t)\})(c_t(x,a) - M_t(x,a))}{\rho_t(x,a) + \gamma}.$$

By $M_t(x,a) \leq c_t(x,a)$, $\rho_t(\cdot) \geq 0$ and $\gamma \geq 0$,

$$\mathbb{E}_{t-1}[\hat{c}_t(x,a)] - \hat{c}_t(x,a) \leq |c_t(x,a) - M_t(x,a)|$$

Thus,

$$\sum_{t=1}^{T} \langle \rho_t, \mathbb{E}_{t-1}[\hat{c}_t] - \hat{c}_t \rangle \leq \sum_{t=1}^{T} \langle \rho_t, \|c_t - M_t\|_1 \rangle.$$

Since $\{\langle \rho_t, \mathbb{E}_{t-1}[\hat{c}_t] - \hat{c}_t \rangle\}_{t=1}^{T}$ is a martingale difference sequence, by using the Azuma–Hoeffding inequality,

$$\Pr \left( \sum_{t=1}^{T} \langle \rho_t, \mathbb{E}_{t-1}[\hat{c}_t] - \hat{c}_t \rangle \geq \epsilon \right) \leq \exp \left( \frac{-\epsilon^2}{2 \sum_{t=1}^{T} \|c_t - M_t\|_1^2} \right) = \delta.$$

Therefore, with probability at least $1 - \delta$, the third term is bounded with

$$\sum_{t=1}^{T} \langle \rho_t, \mathbb{E}_{t-1}[\hat{c}_t] - \hat{c}_t \rangle \leq \sqrt{2 \log \frac{1}{\delta} \sum_{t=1}^{T} \|c_t - M_t\|_1^2}. \tag{25}$$

**Lemma 2.** *Let $\{X_t\}_{t=1}^{T}$ be an $\mathbb{F}$-adapted sequence with the Filtration $\mathbb{F} = (\mathcal{F}_t)_t$. Define $\mathbb{E}_t[\cdot] = \mathbb{E}[\cdot|\mathcal{F}]$. Let $\{\eta_t\}_{t=1}^{T}$ be an $\mathbb{F}$-predictable sequence. Then, if $\eta_t \geq 0$ and $\eta_t(X_t - \mathbb{E}_{t-1}[X_t]) \leq 1.79$, we have*

$$\Pr \left( \sum_{t=1}^{T} \eta_t(X_t - \mu_t) \geq \sum_{t=1}^{T} \eta_t^2 \mathbb{E}_{t-1}[X_t^2] + \log \frac{1}{\delta} \right) \leq \delta. \tag{26}$$

**Proof.** Let $\alpha_t = \mathbb{E}_{t-1}[\eta_t(X_t - \mathbb{E}_{t-1}[X_t])^2] = \eta_t\mathbb{E}_{t-1}[(X_t - \mathbb{E}_{t-1}[X_t])^2]$ and $\mu_t = \mathbb{E}_{t-1}[X_t]$. Since $\eta_t$ is $\mathbb{F}$-predictable, by Markov inequality,

$$
\Pr\left(\sum_{t=1}^T \eta_t(X_t - \mu_t - \alpha_t) \geq \log\frac{1}{\delta}\right) = \Pr\left(\exp\left(\sum_{t=1}^T \eta_t(X_t - \mu_t - \alpha_t)\right) \geq \frac{1}{\delta}\right)
$$
$$
\leq \delta\mathbb{E}\left[\exp\left(\sum_{t=1}^T \eta_t(X_t - \mu_t - \alpha_t)\right)\right]. \tag{27}
$$

Let $Z_n = \exp\left(\sum_{t=1}^n \eta_t(X_t - \mu_t - \alpha_t)\right)$ and $y_{n+1} = \exp(\eta_{n+1}(X_{n+1} - \mu_{n+1} - \alpha_{n+1}))$. Since $Z_{n+1} = Z_n y_{n+1}$ and $Z_n$ is $\mathbb{F}$-adapted,

$$
\mathbb{E}[Z_{n+1}|\mathcal{F}_n] = \mathbb{E}[Z_n y_{n+1}|\mathcal{F}_n]
$$
$$
= Z_n\mathbb{E}[y_{n+1}|\mathcal{F}_n].
$$

By the fact that $\eta_n\alpha_n$ is $\mathbb{F}$-predictable, $\exp(x) \leq 1 + x + x^2$ for $x < 1.79$ and $1 + x \leq \exp(x)$,

$$
\mathbb{E}_{n-1}[y_n] = \exp(-\eta_n\alpha_n)\mathbb{E}_{n-1}[\exp(\eta_n(X_n - \mu_n))]
$$
$$
\leq \exp(-\eta_n\alpha_n)\mathbb{E}_{n-1}[1 + (\eta_n(X_n - \mu_n)) + (\eta_n(X_n - \mu_n))^2]
$$
$$
= \exp(-\eta_n\alpha_n)(1 + \eta_n\mathbb{E}_{n-1}[X_n] - \eta_n\mu_n + \eta_n^2\mathbb{E}_{n-1}[(X_n - \mu_n)^2])
$$
$$
= \exp(-\eta_n\alpha_n)(1 + \eta_n^2\mathbb{E}_{n-1}[(X_n - \mu_n)^2])
$$
$$
\text{(By } \mu_n = \mathbb{E}_{n-1}[X_n])
$$
$$
\leq \exp(-\eta_n\alpha_n)\exp(\eta_n^2\mathbb{E}_{n-1}[(X_n - \mu_n)^2])
$$
$$
= \exp(-\eta_n^2\mathbb{E}_{n-1}[(X_n - \mu_n)^2])\exp(\eta_n^2\mathbb{E}_{n-1}[(X_n - \mu_n)^2]) = 1
$$
$$
\text{(By } \alpha_n = \eta_n\mathbb{E}_{n-1}[(X_n - \mu_n)^2].
$$

Therefore $Z_n$ is a supermartingale: i.e.
$$
\mathbb{E}_n[Z_{n+1}] = Z_n\mathbb{E}_n[y_{n+1}] \leq Z_n.
$$

By tower expectation,
$$
\mathbb{E}[Z_n] = \mathbb{E}\left[\mathbb{E}_{n-1}[Z_n]\right] \leq \mathbb{E}[Z_{n-1}] \leq ... \leq \mathbb{E}[Z_1] = \mathbb{E}[y_1] \leq 1.
$$

Apply this result back to (27),
$$
\Pr\left(\sum_{t=1}^T \eta_t(X_t - \mu_t - \alpha_t) \geq \log\frac{1}{\delta}\right) \leq \delta
$$

Since $\alpha_t = \eta_t\mathbb{E}_{t-1}[(X_t - \mu_t)^2]$ and $\mathbb{E}_{t-1}[(X_t - \mu_t)^2] \leq \mathbb{E}_{t-1}[X_t^2]$,
$$
\Pr\left(\sum_{t=1}^T \eta_t(X_t - \mu_t) \geq \log\frac{1}{\delta} + \sum_{t=1}^T \eta_t^2\mathbb{E}_{t-1}[(X_t - \mu_t)^2]\right) \leq \delta
$$
$$
\Pr\left(\sum_{t=1}^T \eta_t(X_t - \mu_t) \geq \log\frac{1}{\delta} + \sum_{t=1}^T \eta_t^2\mathbb{E}_{t-1}[X_t^2]\right) \leq \delta
$$

∎

To use Lemma 2 for the last term in (11), let $X_t = \sum_{x\in\mathcal{X}_l,a\in\mathcal{A}} \rho(x,a)[\hat{c}_t(x,a) - M_t(x,a)]$ and $\eta_t = \eta = \frac{\gamma}{\|c-M\|_{\mathcal{X}_l}}$, where $\|c - M\|_{\mathcal{X}_l} = \max_{t=1,...,T}\max_{x\in\mathcal{X}_l,a\in\mathcal{A}}|c_t(x,a) - M_t(x,a)|$. Then, by (5),

$$
\mu_t = \mathbb{E}_{t-1}\left[\sum_{x\in\mathcal{X}_l,a\in\mathcal{A}} \rho(x,a)[\hat{c}_t(x,a) - M_t(x,a)]\right]
$$
$$
= \mathbb{E}_{t-1}\left[\sum_{x\in\mathcal{X}_l,a\in\mathcal{A}} \rho(x,a)\frac{c_t(x,a) - M_t(x,a)}{\rho_t(x,a) + \gamma}\mathbb{I}\{(x,a) \in \bar{\mathbf{u}}_L(t)\}\right]
$$
$$
= \sum_{x\in\mathcal{X}_l,a\in\mathcal{A}} \rho(x,a)\frac{\rho_t(x,a)[c_t(x,a) - M_t(x,a)]}{\rho_t(x,a) + \gamma}
$$

By Lemma 2, with probability $1 - \delta'$,

$$\sum_{t=1}^{T} \frac{\gamma}{\|c - M\|_{\mathcal{X}_l}} \left( \sum_{x \in \mathcal{X}_l, a \in \mathcal{A}} \rho(x, a) \left[ \hat{c}_t(x, a) - M_t(x, a) - \frac{\rho_t(x, a)[c_t(x, a) - M_t(x, a)]}{\rho_t(x, a) + \gamma} \right] \right)$$

$$\leq \log \frac{1}{\delta'} + \sum_{t=1}^{T} \left( \frac{\gamma}{\|c - M\|_{\mathcal{X}_l}} \right)^2 \mathbb{E}_{t-1} \left[ \left( \sum_{x \in \mathcal{X}_l, a \in \mathcal{A}} \rho(x, a)[\hat{c}_t(x, a) - M_t(x, a)] \right)^2 \right]$$

By (5),

$$\sum_{t=1}^{T} \frac{\gamma}{\|c - M\|_{\mathcal{X}_l}} \left( \sum_{x \in \mathcal{X}_l, a \in \mathcal{A}} \rho(x, a) \frac{c_t(x, a) - M_t(x, a)}{\rho_t(x, a) + \gamma} [\mathbb{I}\{(x, a) \in \bar{\mathbf{u}}_L(t)\} - \rho_t(x, a)] \right)$$

$$\leq \log \frac{1}{\delta'} + \sum_{t=1}^{T} \left( \frac{\gamma}{\|c - M\|_{\mathcal{X}_l}} \right)^2 \sum_{x \in \mathcal{X}_l, a \in \mathcal{A}} \left( \rho(x, a) \frac{c_t(x, a) - M_t(x, a)}{\rho_t(x, a) + \gamma} \right)^2 \rho_t(x, a)$$

Since $\rho_t(\cdot) \geq 0$ and $\gamma > 0$,

$$\frac{\gamma}{\|c - M\|_{\mathcal{X}_l}} \sum_{t=1}^{T} \sum_{x \in \mathcal{X}_l, a \in \mathcal{A}} \rho(x, a) \frac{c_t(x, a) - M_t(x, a)}{\rho_t(x, a) + \gamma} [\mathbb{I}\{(x, a) \in \bar{\mathbf{u}}_L(t)\} - \rho_t(x, a)]$$

$$\leq \log \frac{1}{\delta'} + \frac{\gamma}{\|c - M\|_{\mathcal{X}_l}} \sum_{t=1}^{T} \sum_{x \in \mathcal{X}_l, a \in \mathcal{A}} \left( \rho(x, a)^2 \frac{[c_t(x, a) - M_t(x, a)]^2}{\rho_t(x, a) + \gamma} \right) \frac{\gamma}{\|c - M\|_{\mathcal{X}_l}}$$

$$\frac{\gamma}{\|c - M\|_{\mathcal{X}_l}} \sum_{t=1}^{T} \sum_{x \in \mathcal{X}_l, a \in \mathcal{A}} \rho(x, a) \frac{c_t(x, a) - M_t(x, a)}{\rho_t(x, a) + \gamma}$$

$$\left[ \mathbb{I}\{(x, a) \in \bar{\mathbf{u}}_L(t)\} - \rho_t(x, a) - \frac{\gamma \rho(x, a)[c_t(x, a) - M_t(x, a)]}{\|c - M\|_{\mathcal{X}_l}} \right]$$

$$\leq \log \frac{1}{\delta'}$$

Since $\rho(x, a), \rho_t(x, a) \geq 0, \gamma > 0$ and $M_t(x, a) \leq c_t(x, a)$, by $\rho(x, a) \leq 1$ and $c_t(x, a) - M_t(x, a) \leq \|c - M\|_{\mathcal{X}_l}$,

$$\frac{\gamma}{\|c - M\|_{\mathcal{X}_l}} \sum_{t=1}^{T} \sum_{x \in \mathcal{X}_l, a \in \mathcal{A}} \rho(x, a) \frac{c_t(x, a) - M_t(x, a)}{\rho_t(x, a) + \gamma} [\mathbb{I}\{(x, a) \in \bar{\mathbf{u}}_L(t)\} - \rho_t(x, a) - \gamma] \leq \log \frac{1}{\delta'}$$

By (5),

$$\frac{\gamma}{\|c - M\|_{\mathcal{X}_l}} \sum_{t=1}^{T} \sum_{x \in \mathcal{X}_l, a \in \mathcal{A}} \rho(x, a)[\hat{c}_t(x, a) - c_t(x, a)] \leq \log \frac{1}{\delta}$$

For each layer $l$, with probability $1 - \delta'$,

$$\sum_{t=1}^{T} \sum_{x \in \mathcal{X}_l, a \in \mathcal{A}} \rho(x, a)[\hat{c}_t(x, a) - c_t(x, a)] \leq \frac{\|c - M\|_{\mathcal{X}_l}}{\gamma} \log \frac{1}{\delta'}$$

By union bound on all layers, with probability $1 - L\delta'$,

$$\sum_{t=1}^{T}\sum_{x,a}\rho(x,a)[\hat{c}_t(x,a) - c_t(x,a)] \leq \sum_{l=1}^{L}\frac{\|c - M\|_{\mathcal{X}_l}}{\gamma}\log\frac{1}{\delta'}$$

$$\leq \frac{L}{\gamma}\log\frac{1}{\delta'}\max_{l=1,\ldots,L}\|c - M\|_{\mathcal{X}_l}$$

$$= \frac{L}{\gamma}\log\frac{1}{\delta'}\max_{t=1,\ldots,T}\|c_t - M_t\|_{\infty}$$

Setting $\delta' = \delta/L$, with probability $1 - \delta$, the last term in (11) is bounded as

$$\sum_{t=1}^{T}\langle\rho^*, \hat{c}_t - c_t\rangle \leq \frac{L}{\gamma}\log\frac{L}{\delta}\max_{t=1,\ldots,T}\|c_t - M_t\|_{\infty}. \tag{28}$$

Applying (23), (24), (25) and (28) back to (11),

$$\mathcal{R}_T(\rho^*, \{c_t\}_{t=1}^T) \leq \frac{L}{\eta}\log\frac{|\mathcal{X}||\mathcal{A}|}{L} + \frac{\eta}{2\gamma^2}\sum_{t=1}^{T}\|c_t - M_t\|_{\infty}^2 + \gamma\sum_{t=1}^{T}\|c_t - M_t\|_1$$

$$+ \sqrt{2\log\frac{1}{\delta}\sum_{t=1}^{T}\|c_t - M_t\|_1^2} + \frac{L}{\gamma}\log\frac{L}{\delta}\max_{t=1,\ldots,T}\|c_t - M_t\|_{\infty}$$

$$\leq \frac{L}{\eta}\log\frac{|\mathcal{X}||\mathcal{A}|}{L} + \frac{\eta}{2\gamma^2}\sum_{t=1}^{T}\|c_t - M_t\|_{\infty}^2 + \gamma\sum_{t=1}^{T}\|c_t - M_t\|_1$$

$$+ \sqrt{2\log\frac{1}{\delta}\sum_{t=1}^{T}\|c_t - M_t\|_1^2} + \frac{L}{\eta}\log\frac{L}{\delta}\max_{t=1,\ldots,T}\|c_t - M_t\|_{\infty}$$

$$\text{(Since } \eta \leq \gamma \text{ if } \gamma = \eta^{1/3} \text{ and } \eta, \gamma \leq 1)$$

Let $\eta = \left(\dfrac{L\log\frac{|\mathcal{X}||\mathcal{A}|}{L} + L\log\frac{L}{\delta}\max_t\|c_t - M_t\|_{\infty}}{\sum_{t=1}^{T}\frac{\|c_t - M_t\|_{\infty}^2}{2} + \|c_t - M_t\|_1}\right)^{3/4}$ and $\gamma = \eta^{1/3}$.

$$\mathcal{R}_T(\rho^*, \{c_t\}_{t=1}^T) \leq \left(L\log\frac{|\mathcal{X}||\mathcal{A}|}{L} + L\log\frac{L}{\delta}\max_t\|c_t - M_t\|_{\infty}\right)^{1/4}$$

$$\cdot\left(\sum_{t=1}^{T}\frac{\|c_t - M_t\|_{\infty}^2}{2} + \|c_t - M_t\|_1\right)^{3/4}$$

$$+ \sqrt{2\log\frac{1}{\delta}\sum_{t=1}^{T}\|c_t - M_t\|_1^2}$$

∎

## A.5 PROOF OF THEOREM 4

The expected total regret can be decomposed into local regrets of each phase as below, where $N > 1$ denotes the number of phases that $T$ episodes are broken into.

$$\mathbb{E}\left[\sum_{t=1}^{T}\langle\rho_t - \rho^*, c_t\rangle\right] = \mathbb{E}\left[\sum_{i=1}^{N}\sum_{t=s_i}^{s_{i+1}-1}\langle\rho_t - \rho^*, c_t\rangle\right]$$

Since the randomness of the future does not affect the past,

$$\mathbb{E}\left[\sum_{t=1}^{T}\langle\rho_t - \rho^*, c_t\rangle\right] = \sum_{i=1}^{N}\mathbb{E}_{\mathbf{u}(s_{i+1}-1)}\left[\sum_{t=s_i}^{s_{i+1}-1}\langle\rho_t, c_t\rangle - \sum_{t=s_i}^{s_{i+1}-1}\langle\rho^*, c_t\rangle\right]$$

$$\leq \sum_{i=1}^{N}\mathbb{E}_{\mathbf{u}(s_{i+1}-1)}\left[\sum_{t=s_i}^{s_{i+1}-1}\langle\rho_t, c_t\rangle - \min_{\rho\in\Delta(\mathcal{M})}\sum_{t=s_i}^{s_{i+1}-1}\langle\rho, c_t\rangle\right]. \tag{29}$$

Now we can consider solving for the upper bound as the problem of each phase independently, where the local regret of each phase is expressed with respect to its local optimum: $\rho_i^* = \arg\min_{\rho\in\Delta(\mathcal{M})}\sum_{t=s_i}^{s_{i+1}-1}\langle\rho, \hat{c}_t\rangle$. If $\rho_{t+1}$ is the solution of (6) with $\eta = \eta_i$ and $t = s_i, \ldots, s_{i+1} - 1$, by Theorem 2,

$$\mathbb{E}\left[\sum_{t=s_i}^{s_{i+1}-1}\langle\rho_t - \rho_i^*, c_t\rangle\right] \leq \frac{L}{\eta_i}\log\frac{|\mathcal{X}||\mathcal{A}|}{L} + \frac{\eta_i}{2\gamma_i}\sum_{t=s_i}^{s_{i+1}-1}\|c_t - M_t\|_2^2 + \sum_{t=s_i}^{s_{i+1}-1}\gamma_i\|c_t - M_t\|_1.$$

Note that the term $\frac{L}{\eta_i}\log\frac{|\mathcal{X}||\mathcal{A}|}{L}$ represents the initial suboptimality assuming that $\rho_{s_i}$ is initialized as the uniform distribution. However, the logic behind regularizing the Bregman divergence (between the current and past occupancy measures) is that the occupancy measure learned in an episode will suffer a lower cost in the next episode than random initialization. Therefore the bound conservatively holds for $\rho_{s_i}$ learned from the previous phase instead of initializing it every phase.

By Algorithm 2, use $\gamma_i = \sqrt{\eta_i}$.

$$\mathbb{E}\left[\sum_{t=s_i}^{s_{i+1}-1}\langle\rho_t - \rho_i^*, c_t\rangle\right] \leq \frac{L}{\eta_i}\log\frac{|\mathcal{X}||\mathcal{A}|}{L} + \frac{\eta_i}{2\gamma_i}\sum_{t=s_i}^{s_{i+1}-1}\|c_t - M_t\|_2^2 + \sum_{t=s_i}^{s_{i+1}-1}\gamma_i\|c_t - M_t\|_1$$

$$= \frac{L}{\eta_i}\log\frac{|\mathcal{X}||\mathcal{A}|}{L} + \sqrt{\eta_i}\sum_{t=s_i}^{s_{i+1}-1}\left\{\frac{1}{2}\|c_t - M_t\|_2^2 + \|c_t - M_t\|_1\right\}.$$

Since $\mathbb{E}\left[|\bar{c}_t(x,a) - M_t(x,a)|\right] = |c_t(x,a) - M_t(x,a)|$ with respect to the randomness of the trajectory $\bar{\mathbf{u}}_L(t)$,

$$\mathbb{E}\left[\sum_{t=s_i}^{s_{i+1}-1}\langle\rho_t - \rho_i^*, c_t\rangle\right] \leq \frac{L}{\eta_i}\log\frac{|\mathcal{X}||\mathcal{A}|}{L} + \sqrt{\eta_i}\mathbb{E}\left[\sum_{t=s_i}^{s_{i+1}-1}\left\{\frac{1}{2}\|\bar{c}_t - M_t\|_2^2 + \|\bar{c}_t - M_t\|_1\right\}\right].$$

By step 4 of Algorithm 2, $\eta_i^{-1}D_0 \geq \sqrt{\eta_i}\Psi_{s_i:s_{i+1}-1}$ for all $i = 1, \ldots, N$.

$$\mathbb{E}\left[\sum_{t=s_i}^{s_{i+1}-1}\langle\rho_t - \rho_i^*, c_t\rangle\right] \leq \frac{2L}{\eta_i}\log\frac{|\mathcal{X}||\mathcal{A}|}{L}$$

Adding over $N$ phases, by Equation (29),

$$\mathbb{E}\left[\sum_{t=1}^{T}\langle\rho_t - \rho^*, c_t\rangle\right] = L\log\frac{|\mathcal{X}||\mathcal{A}|}{L}\left(2\sum_{i=1}^{N}\frac{1}{\eta_i}\right) = D_0\left(2\sum_{i=1}^{N}\frac{1}{\eta_i}\right).$$

From $\sum_{i=1}^{N}(1/2)^i \leq 1$ and $\eta_{N-1} = \eta_0/2^{N-1}$,

$$2\sum_{i=1}^{N}\frac{1}{\eta_i} = \frac{2}{\eta_0}\sum_{i=1}^{N}2^i = \frac{2}{\eta_0}2^{N+1}\sum_{i=1}^{N}2^{i-N-1} = \frac{2^{N+2}}{\eta_0}\sum_{i=1}^{N}(\frac{1}{2})^i \leq \frac{2^{N+2}}{\eta_0} = \frac{8}{\eta_{N-1}}. \tag{30}$$

By $\sqrt{\eta_{N-1}}\Psi_{s_{N-1}:s_N} > \eta_{N-1}^{-1}D_0 \geq \sqrt{\eta_{N-1}}\Psi_{s_{N-1}:s_{N-1}}$ and the monotonicity of $\Psi$,

$$\eta_{N-1}^{-3/2} < \frac{\Psi_{s_{N-1}:s_N}}{D_0} \leq \frac{\Psi_{1:T}}{D_0}.$$

Then,

$$\mathbb{E}[\mathcal{R}_T(\rho^*, \{c_t\}_{t=1}^T)] \leq D_0 \frac{8}{\eta_{N-1}}$$

$$< 8D_0 \left(\frac{\Psi_{1:T}}{D_0}\right)^{2/3}$$

$$= 8D_0^{1/3}\Psi_{1:T}^{2/3}$$

$$= 8 \left(L \log \frac{|\mathcal{X}||\mathcal{A}|}{L}\right)^{1/3} \left(\sum_{t=1}^{T} \frac{1}{2}\|\bar{c}_t - M_t\|_2^2 + \|\bar{c}_t - M_t\|_1\right)^{2/3}.$$

Taking expectation over the randomness of the trajectory $\bar{\mathbf{u}}_L(T)$,

$$\mathbb{E}[\mathcal{R}_T(\rho^*, \{c_t\}_{t=1}^T)] \leq 8 \left(L \log \frac{|\mathcal{X}||\mathcal{A}|}{L}\right)^{1/3} \left(\sum_{t=1}^{T} \frac{1}{2}\|c_t - M_t\|_2^2 + \|c_t - M_t\|_1\right)^{2/3}.$$

Note that determining $\eta_i^{-1}D_0 < \sqrt{\eta_i}\Psi_{s_i:t}$ for each episode $t$ does not require additional suffering of cost. As in Algorithm 2, determining $\eta_i$ and $\gamma_i$ can come after the entire rollout of episode $t$, as they are only needed for computing $\hat{c}_t$, $M_{t+1}$ and $\rho_{t+1}$. Therefore this is the final bound unlike Lemma 12 of Rakhlin and Sridharan [2013], which suffers additional cost for finding $\Psi$. ∎

## A.6 PROOF OF THEOREM 5

The proof is similar to Appendix A.5 but simpler. Again, the total regret can be decomposed into local regrets of each phase.

$$\sum_{t=1}^{T}\langle c_t, \rho_t - \rho^*\rangle = \sum_{i=1}^{N}\sum_{t=s_i}^{s_{i+1}-1}\langle \rho_t - \rho^*, c_t\rangle$$

$$\leq \sum_{i=1}^{N}\sum_{t=s_i}^{s_{i+1}-1}\langle \rho_t, c_t\rangle - \min_{\rho \in \Delta(\mathcal{M})}\sum_{t=s_i}^{s_{i+1}-1}\langle \rho, c_t\rangle$$

Now the upper bound is the problem of optimizing (6) with $\eta = \eta_i$ and $t = s_i, \ldots, s_{i+1} - 1$. By Theorem 1,

$$\sum_{t=s_i}^{s_{i+1}-1}\langle \rho_t - \rho_i^*, c_t\rangle \leq \frac{L}{\eta_i}\log\frac{|\mathcal{X}||\mathcal{A}|}{L} + \sum_{t=1}^{T}\frac{\eta_i}{2}\|c_t - M_t\|_\infty^2$$

where $\rho_i^* = \arg\min_{\rho \in \Delta(\mathcal{M})}\sum_{t=s_i}^{s_{i+1}-1}\langle \rho, c_t\rangle$.

By $\eta_i^{-1}D_0 \geq \eta_i\Psi_{s_i:s_{i+1}-1}$ according to our doubling trick algorithm,

$$\sum_{t=s_i}^{s_{i+1}-1}\langle \rho_t - \rho_i^*, c_t\rangle \leq \frac{2L}{\eta_i}\log\frac{|\mathcal{X}||\mathcal{A}|}{L}.$$

Adding over $N$ phases,

$$\sum_{t=1}^{T}\langle c_t, \rho_t - \rho^*\rangle \leq L\log\frac{|\mathcal{X}||\mathcal{A}|}{L}\left(2\sum_{i=1}^{N}\frac{1}{\eta_i}\right)$$

By (30), $\eta_{N-1}^{-1} D_0 < \eta_{N-1} \Psi_{s_{N-1}:s_N}$ and the monotonicity of $\Psi$,

$$
\begin{aligned}
\mathcal{R}_T(\rho^*, \{c_t\}_{t=1}^T) &\le D_0 \frac{8}{\eta_{N-1}} \\
&< 8 D_0 \left( \frac{\Psi_{1:T}}{D_0} \right)^{1/2} \\
&= 8 D_0^{1/2} \Psi_{1:T}^{1/2} \\
&= 8 \sqrt{ \frac{L}{2} \log \frac{|\mathcal{X}||\mathcal{A}|}{L} \sum_{t=1}^T \|c_t - M_t\|_\infty^2 }
\end{aligned}
$$

■

## A.7 PROOF OF THEOREM 6

Since the constraint set of occupancy measure is unknown, the regret of OREPS-OPIX under unknown transition setting can be decomposed as (31). Note the additional error term $\rho_t - \hat{\rho}_t$ as opposed to (11) used for the analysis of Theorem 3.

$$
\mathcal{R}_T(\rho^*, \{c_t\}_{t=1}^T) = \sum_{t=1}^T \left[ \langle \rho_t - \hat{\rho}_t, c_t \rangle + \langle \hat{\rho}_t, c_t - \hat{c}_t \rangle + \langle \hat{\rho}_t - \rho^*, \hat{c}_t \rangle + \langle \rho^*, \hat{c}_t - c_t \rangle \right]. \tag{31}
$$

**Lemma 3** (Lemma 5 of Jin et al. [2020]). *With probability at least $1 - 6\delta$, for $\hat{\rho}_t$ estimated with $\rho^{P,\pi_t}$ under transition probability $P \in \mathcal{P}$, where the confidence set $\mathcal{P}$ is defined as (14),*

$$
\sum_{t=1}^T \langle \rho_t - \hat{\rho}_t, c_t \rangle = \mathcal{O}\left( L|\mathcal{X}| \sqrt{ |\mathcal{A}| T \log \left( \frac{T|\mathcal{X}||\mathcal{A}|}{\delta} \right) } \right).
$$

By Lemma 3, with probability at least $1 - 6\delta$, the first term is bounded as

$$
\sum_{t=1}^T \langle \rho_t - \hat{\rho}_t, c_t \rangle = \mathcal{O}\left( L|\mathcal{X}| \sqrt{ |\mathcal{A}| T \log \left( \frac{T|\mathcal{X}||\mathcal{A}|}{\delta} \right) } \right). \tag{32}
$$

The second term can be decomposed further as

$$
\sum_{t=1}^T \langle \hat{\rho}_t, c_t - \hat{c}_t \rangle = \sum_{t=1}^T \langle \hat{\rho}_t, c_t - \mathbb{E}_{t-1}[\hat{c}_t] \rangle + \sum_{t=1}^T \langle \hat{\rho}_t, \mathbb{E}_{t-1}[\hat{c}_t] - \hat{c}_t \rangle.
$$

From the definition of our cost estimator with the upper confidence bound,

$$
\begin{aligned}
\sum_{t=1}^T \langle \hat{\rho}_t, c_t - \mathbb{E}_{t-1}[\hat{c}_t] \rangle &= \sum_{t=1}^T \sum_{x,a} \hat{\rho}_t(x,a) \left[ c_t(x,a) - \frac{c_t(x,a) - M_t(x,a)}{u_t(x,a) + \gamma} \rho_t(x,a) - M_t(s,a) \right] \\
&= \sum_{t=1}^T \sum_{x,a} \frac{\hat{\rho}_t(x,a)(c_t(x,a) - M_t(x,a))}{u_t(x,a) + \gamma}(u_t(x,a) + \gamma - \rho_t(x,a)).
\end{aligned}
$$

By $M_t(x,a) \le c_t(x,a) \le M_t(x,a) + 1$, $\hat{\rho}_t(\cdot) \ge 0$ and $u_t(x,a) \ge \hat{\rho}_t(x,a)$,

$$
\begin{aligned}
\sum_{t=1}^T \langle \hat{\rho}_t, c_t - \mathbb{E}_{t-1}[\hat{c}_t] \rangle &\le \sum_{t=1}^T \sum_{x,a} (c_t(x,a) - M_t(x,a))(u_t(x,a) + \gamma - \rho_t(x,a)) \\
&\le \sum_{t=1}^T \sum_{x,a} |u_t(x,a) - \rho_t(x,a)| + (c_t(x,a) - M_t(x,a))\gamma \\
&= \sum_{t=1}^T \sum_{x,a} |u_t(x,a) - \rho_t(x,a)| + \gamma \sum_{t=1}^T \|c_t - M_t\|_1.
\end{aligned}
$$

**Lemma 4** (Lemma 4 of Jin et al. [2020]). *With probability at least $1 - 6\delta$, for transition functions $P_t^x \in \mathcal{P}$ for all states $x \in \mathcal{X}$, where the confidence set $\mathcal{P}$ is defined as (14), the cumulative error of occupancy measure with respect to $\rho_t$ of known transition setting is bouned as*

$$\sum_{t=1}^{T} \sum_{x \in \mathcal{X}, a \in \mathcal{A}} \left| \rho^{P_t^x, \pi_t}(x, a) - \rho_t(x, a) \right| \leq \mathcal{O}\left( L|\mathcal{X}| \sqrt{|\mathcal{A}| T \log\left( \frac{T|\mathcal{X}||\mathcal{A}|}{\delta} \right)} \right)$$

Since $u_t(x, a) = \max_{P \in \mathcal{P}} \rho^{P, \pi_t}(x, a)$, by Lemma 4, with probability at least $1 - 6\delta$,

$$\sum_{t=1}^{T} \langle \hat{\rho}_t, c_t - \mathbb{E}_{t-1}[\hat{c}_t] \rangle \leq O\left( L|\mathcal{X}| \sqrt{|\mathcal{A}| T \log\left( \frac{T|\mathcal{X}||\mathcal{A}|}{\delta} \right)} \right) + \gamma \sum_{t=1}^{T} \|c_t - M_t\|_1$$

Since $\{\langle \hat{\rho}_t, \mathbb{E}_{t-1}[\hat{c}_t] - \hat{c}_t \rangle\}_{t=1}^{T}$ is a martingale difference sequence, by using the Azuma–Hoeffding inequality, with probability at least $1 - \delta$,

$$\sum_{t=1}^{T} \langle \hat{\rho}_t, \mathbb{E}_{t-1}[\hat{c}_t] - \hat{c}_t \rangle \leq \sqrt{2 \log \frac{1}{\delta} \sum_{t=1}^{T} \|c_t - M_t\|_1^2}.$$

With probability at least $1 - 7\delta$, the second term is bounded as

$$\sum_{t=1}^{T} \langle \hat{\rho}_t, c_t - \hat{c}_t \rangle \leq \mathcal{O}\left( L|\mathcal{X}| \sqrt{|\mathcal{A}| T \log\left( \frac{T|\mathcal{X}||\mathcal{A}|}{\delta} \right)} + \sqrt{\log \frac{1}{\delta} \sum_{t=1}^{T} \|c_t - M_t\|_1^2} \right) + \gamma \sum_{t=1}^{T} \|c_t - M_t\|_1 \quad (33)$$

Since $\hat{\rho}_t$ optimizes for $\hat{c}_t$, from the analyses of Theorem 1 and 3, the third term is bounded as

$$\sum_{t=1}^{T} \langle \hat{\rho}_t - \rho^*, \hat{c}_t \rangle \leq \frac{L}{\eta} \log \frac{|\mathcal{X}||\mathcal{A}|}{L} + \sum_{t=1}^{T} \frac{\eta}{2} \|\hat{c}_t - M_t\|_\infty^2$$

$$\leq \frac{L}{\eta} \log \frac{|\mathcal{X}||\mathcal{A}|}{L} + \frac{\eta}{2\gamma^2} \sum_{t=1}^{T} \|c_t - M_t\|_\infty^2. \quad (34)$$

Since $u_t(x, a) \geq \rho_t(x, a)$, from the analysis of Theorem 3 using Lemma 2, the fourth term is bounded as

$$\sum_{t=1}^{T} \langle \rho^*, \hat{c}_t - c_t \rangle \leq \frac{L}{\gamma} \log \frac{L}{\delta} \max_{t=1,\ldots,T} \|c_t - M_t\|_\infty. \quad (35)$$

Finally, applying 32, 33, 34 and 35 back to 31 and letting $\gamma = \eta^{1/3}$, with probability at least $1 - 7\delta$,

$$\mathcal{R}_T(\rho^*, \{c_t\}_{t=1}^{T}) \leq \mathcal{O}\left( L|\mathcal{X}| \sqrt{|\mathcal{A}| T \log\left( \frac{T|\mathcal{X}||\mathcal{A}|}{\delta} \right)} + \sqrt{\log \frac{1}{\delta} \sum_{t=1}^{T} \|c_t - M_t\|_1^2} \right)$$

$$+ \eta^{1/3} \sum_{t=1}^{T} \left[ \frac{\|c_t - M_t\|_\infty^2}{2} + \|c_t - M_t\|_1 \right]$$

$$+ \frac{L}{\eta} \left[ \log \frac{|\mathcal{X}||\mathcal{A}|}{L} + \log \frac{L}{\delta} \max_{t=1,\ldots,T} \|c_t - M_t\|_\infty \right]$$

Let $\eta = \left( \dfrac{L \log \frac{|\mathcal{X}||\mathcal{A}|}{L} + L \log \frac{L}{\delta} \max_t \|c_t - M_t\|_\infty}{\sum_{t=1}^T \frac{\|c_t - M_t\|_\infty^2}{2} + \|c_t - M_t\|_1} \right)^{3/4}$,

$$
\mathcal{R}_T(\rho^*, \{c_t\}_{t=1}^T) \leq \left( L \log \frac{|\mathcal{X}||\mathcal{A}|}{L} + L \log \frac{L}{\delta} \max_t \|c_t - M_t\|_\infty \right)^{1/4}
$$
$$
\cdot \left( \sum_{t=1}^T \frac{\|c_t - M_t\|_\infty^2}{2} + \|c_t - M_t\|_1 \right)^{3/4}
$$
$$
+ \mathcal{O} \left( L|\mathcal{X}| \sqrt{|\mathcal{A}|T \log \left( \frac{T|\mathcal{X}||\mathcal{A}|}{\delta} \right)} + \sqrt{\log \frac{1}{\delta} \sum_{t=1}^T \|c_t - M_t\|_1^2} \right)
$$

# B  EXPERIMENTS

## B.1  EXPERIMENTAL DETAILS

We provide the details of the experiment in Section 6 as Table 1. Additionally, we specify $t_m = 1000$ episodes between the change of obstacle locations for better predictability. Also, agent's starting location was randomly assigned at the beginning of each episode and the goal location was fixed across episodes. And all three obstacles moved randomly every $t_m$ episodes, but in a restricted manner so that they do not obstruct the way from the starting point to the goal: that is, there is always a way from the start to the goal without encountering any obstacles. Lastly, the experiment was repeated ten times and the mean and variance of ten repetitions are shown in Figure 1.

Table 1: Parameters used in the experiments

| Parameter | Description | Value |
|---|---|---|
| $\epsilon$ | Default cost | 0.01 |
| $L$ | Timeout (number of layers) | 200 |
| $\eta_{\text{OREPS}}$ | Learning rate for OREPS and OREPS-IX | $2.1 \times 10^{-3}$ |
| $\eta_{\text{OREPS-OPIX}}$ | Learning rate for OREPS-OPIX | 0.2 |

From Zimin and Neu [2013], the learning rate for OREPS and OREPS-IX was determined as $\eta_{\text{OREPS}} = \sqrt{L \frac{\log \frac{|\mathcal{X}||\mathcal{A}|}{L}}{T|\mathcal{X}||\mathcal{A}|}}$. However, since the perfect predictor we used for OREPS-OPIX has zero error for cost estimation, we can set an arbitrarily high learning rate as long as it is less than 1 (for the high probability guarantee in Theorem 3). After a sparse exploration of parameters, we chose $\eta_{\text{OREPS-OPIX}} = 0.2$. With a higher learning rate, the algorithm converges even faster at the cost of higher variance. And the same learning rate was used for OREPS-OPIX with latest predictors.