# OpenReview forum: "Optimistic Regret Bounds for Online Learning in Adversarial Markov Decision Processes"
_auai.org/UAI/2024/Conference — UAI 2024 oral_

### Official Review · Reviewer_QVM8 · 2024-03-21

**Q2-1 Originality-Novelty:** 3
**Q2-2 Correctness-Technical Quality:** 4
**Q2-5 Clarity Of Writing:** 4

**Q10 Ethical Concerns:**

No.

**Q1 Summary And Contributions:**

This paper studies the problem of online learning in Adversarial Markov Decision Process (AMDP). The main contribution of the authors is the introduction of a novel optimistically biased cost estimator and a corresponding algorithm for which they could establish optimistic regret bounds, both in expectation and in high probability.

They authors showed that their estimator benefits from smaller variance than the implicit exploration estimator (IX). They also show how to derive an algorithm with this estimator using a mirror-descent update, called OREPS-OPIX. They showed how to extend the algorithm to anytime setting using a doubling trick and could establish comparable expected regret bounds. Finally, they provide numerical experiment that illustrate the improved variance compared to OREPRS-IX and confirm their optimistic prediction assumption.

**Q2-3 Extent To Which Claims Are Supported By Evidence:**

4: Excellent: all claims are supported by very convincing evidence (in the form of comprehensive experimental evaluation, rigorous mathematical proofs, detailed (pseudo-)code, precise references, well-motivated and realistic assumptions) and the authors deliver what they promise.

**Q2-4 Reproducibility:**

3: Good: key resources (e.g. proofs, code, data) are available and key details (e.g. proofs, experimental setup) are sufficiently well-described for competent researchers to confidently reproduce the main results.

**Q3 Main Strengths:**

There are several strengths in the paper. The two main ones in my opinion are the excellent writing of the paper, with a great structure, clear notations, and detailed literature review. The second main strength of this work is the quality of the theoretical results and the completeness of the work, covering from full-information to worst-case, from expectation to high-probability bounds, as well as an anytime theoretical analysis and numerical experiments.

**Q4 Main Weakness:**

I did not find weaknesses in the present work.

**Q5 Detailed Comments To The Authors:**

I enjoyed reading and reviewing this paper.

**Q9 Complying With Reviewing Instructions:**

Yes

---

> ### Author Rebuttal · Authors · 2024-04-07
>
> We thank you for your positive feedback. The code associated with the experiment section is made public at "https://anonymous.4open.science/r/OREPS-OPIX-651A". We invite you to review and execute the code as needed.

---

### Official Review · Reviewer_RMx6 · 2024-03-22

**Q2-1 Originality-Novelty:** 3
**Q2-2 Correctness-Technical Quality:** 3
**Q2-5 Clarity Of Writing:** 3

**Q1 Summary And Contributions:**

This paper considers the AMDP with time-varying cost functions derives the worst-case regret bound for the full-information feedback setting and the bandit feedback setting. When developing the theoretical upper bound, this work develops a new cost estimator.

**Q2-3 Extent To Which Claims Are Supported By Evidence:**

3: Good: the main claims are supported by convincing evidence (in the form of adequate experimental evaluation, proofs, (pseudo-)code, references, assumptions).

**Q2-4 Reproducibility:**

3: Good: key resources (e.g. proofs, code, data) are available and key details (e.g. proofs, experimental setup) are sufficiently well-described for competent researchers to confidently reproduce the main results.

**Q3 Main Strengths:**

This paper considers the AMDPs setting, which is obviously more challenging than the traditional MDPs. The derived bound matches the best-known result for perfect estimation and can be generalized to futile estimation. So, I believe this work has its own specific novelty and made significant contributions. This work is also complete and makes adequate discussions on its extension to unknown transiton and the anytime setting.

**Q4 Main Weakness:**

The classical MDPs are a special case of AMDPs. But the connection among results for AMDPs and results for MDPs is not clear.

**Q5 Detailed Comments To The Authors:**

1. When all costs are fixed (w.r.t. the time), are results dereived in the paper degenerate to some existing results known in MDPs?
2. In the numerical experiments, are all cost functions same? Which parts of cost function will depend on the time $t$?

**Q9 Complying With Reviewing Instructions:**

Yes

---

> ### Author Rebuttal · Authors · 2024-04-07
>
> We thank you for your insightful review. We summarized our response in two points.
>
> Connection to MDP: If the cost function remains constant over time, direct optimization of the cost function without bounding the relative entropy becomes feasible. In the case of full information feedback, the cost function is fully observed after the initial episode, resulting in zero regret from the second episode onward. In bandit feedback case, we have a bound with diminishing prediction error $c_t-M_t$, as costs are revealed for additional states and actions. The rate at which the error reduces and efficient strategies for its reduction present an interesting direction for future research. We will mention this point in the final version of our paper.
>
> Cost function in numerical experiments: In the experiment section, the cost function depends on the next state as a result of the current state and action. That is $c(x,a)=c(x')$, where $x'=T(x,a)$ and $T$ is a deterministic state transition function. The cost $c(x^*)$ associated with the goal state $x^*$ remains constant, while the cost of other states varies based on the neighboring states. Specifically, with probability $n/4$, $c(x')=1$ and $c(x')=\epsilon$ otherwise. Here $n$ depends on the number of neighboring states in turbulence and whether the state itself is already turbulent.

---

### Official Review · Reviewer_U8o8 · 2024-03-23

**Q2-1 Originality-Novelty:** 3
**Q2-2 Correctness-Technical Quality:** 3
**Q2-5 Clarity Of Writing:** 3

**Q1 Summary And Contributions:**

The paper studies adversarial MDP and proposes OREPS-OPIX, which can obtain a sublinear result depending on the changes in the cost function. The result is thus less pessimistic than most algorithms for adversarial MDP.

**Q2-3 Extent To Which Claims Are Supported By Evidence:**

3: Good: the main claims are supported by convincing evidence (in the form of adequate experimental evaluation, proofs, (pseudo-)code, references, assumptions).

**Q2-4 Reproducibility:**

3: Good: key resources (e.g. proofs, code, data) are available and key details (e.g. proofs, experimental setup) are sufficiently well-described for competent researchers to confidently reproduce the main results.

**Q3 Main Strengths:**

The idea of using optimistic learning in adversarial MDP is very interesting. To my knowledge, optimistic learning is often used in learning in games. Similar to the idea of this paper, it is found that it can accommodate a range of changes in cost (not necessarily the worst case).  The results extends to uncertain transitions, and bandit feedback. The algorithm is also evaluated empirically and is demonstrated to be efficient.

**Q4 Main Weakness:**

Please see Q5 for the detailed questions.

**Q5 Detailed Comments To The Authors:**

1. Does optimistic learning enjoy bounds such as RVU property in AMDP (ref [1])?
2. Can you provide some examples of how one may design M_t?

[1] Syrgkanis, V., Agarwal, A., Luo, H., & Schapire, R. E. (2015). Fast convergence of regularized learning in games. Advances in Neural Information Processing Systems, 28.

**Q9 Complying With Reviewing Instructions:**

Yes

---

> ### Author Rebuttal · Authors · 2024-04-07
>
> We are grateful for your insightful review. We summarized our response in two points as below.
>
> Regret bounded by Variation in Utilities (RVU) property: Syrgkanis et al. (2015) established that Optimistic Mirror Descent (as introduced by Rakhlin and Sridharan (2013); Chiang et al. (2012)) satisfies the RVU property. However, our approach relies on a single-projection method suggested by Joulani et al. (2017) and the analysis loses the term with $||\rho_t-\rho_{t+1}||_1^2$ in the full information setting.
>
> Design of cost predictor $M_t$: As used in the experiment, a predictor may predict a value based on the latest experience from a state-action pair, and reset or discount the value if staled. In practice, adequate discounting  suffices to uphold the optimistic prediction assumption in terms of the sum over all states and actions.

---

### Official Review · Reviewer_M3KM · 2024-03-23

**Q2-1 Originality-Novelty:** 3
**Q2-2 Correctness-Technical Quality:** 3
**Q2-5 Clarity Of Writing:** 3

**Q1 Summary And Contributions:**

An interesting paper on Adversarial MDP to overcome the current limitations of the existing AMDP methods that consider pessimistic regret analysis. This paper proposes a variant of AMDP to overcome this limitation by considering to minimize the regret while using a set of cost predictors. The method proposed consists of a policy search method that achieves a regret bound based on optimistic regret with high probability. This bound degrades with better estimation of the cost predictors.

**Q2-3 Extent To Which Claims Are Supported By Evidence:**

3: Good: the main claims are supported by convincing evidence (in the form of adequate experimental evaluation, proofs, (pseudo-)code, references, assumptions).

**Q2-4 Reproducibility:**

3: Good: key resources (e.g. proofs, code, data) are available and key details (e.g. proofs, experimental setup) are sufficiently well-described for competent researchers to confidently reproduce the main results.

**Q3 Main Strengths:**

The contributions of the paper consists of
-A worst-case regret bound using a cumulative estimation error  of the cost predictors ;
- New estimators that has variance reduction benefits  and
- An anytime extension for continuous training and with regret guarantees.

The formalisation is accurate, the claims are well supported numerically.

**Q4 Main Weakness:**

Some proposed methods are not well motivated as equation(1) while an MC method could also be applied. How equation (5) could be related to an update value in Q-learning.

**Q5 Detailed Comments To The Authors:**

The behavior control is dictated by an occupancy measure while an MC method could also be considered. A justification of this choice is welcome.
A discussion on Equation 5 and the update value in Q-learning is also welcome.

In general in adversarial decision-making problems when formalized as a stochastic game with a leader, the solution concept is the (strong or weak)  Stackelberg equilibrium. Did you characterize the solution of your method from the Stackelberg equilibrium point of view.

**Q9 Complying With Reviewing Instructions:**

Yes

---

> ### Author Rebuttal · Authors · 2024-04-07
>
> We thank you for your insightful review. We summarized our response in three points.
>
> MC method for control: Equation (1) defines a control policy proportional to the value of the occupancy measure, which is optimized for the cost estimator in equation (5) and regularized by KL divergence. On the other hand, the Monte Carlo method controls the behavior based on average value. In this case, an action would be selected according to the average cost incurred by each action at a given state. While average cost is valuable for constructing a cost predictor, it exhibits sluggish adaptation to dynamic cost functions that evolve over time. Thus, it is impractical to directly use it as a behavior policy.
>
> Cost estimator and Q-learning: Equation (5) defines an optimistic cost estimator that incorporates the cost suffered from the most recent visit as well as the optimistic cost predictor. Unlike Q-learning, which iteratively updates the value (or expected cumulative cost), equation (5) provides a one-shot estimate of single-step costs. An interesting research direction involves integrating the optimistic cost predictor into Q-learning. Such an extension would enable updating the Q-values of states and actions that are not encountered during the current episode.
>
> Stackelberg equilibrium: We can formalize the problem as a stochastic game, where a leader selects the occupancy measure and a follower chooses the cost function. However, a fundamental assumption underlying our work is that the choice of the cost function remains oblivious and does not commit to a best response. Nonetheless, an interesting research direction lies in extending our analysis to Stackelberg games.

---

### Official Review · Reviewer_sFYU · 2024-03-30

**Q2-1 Originality-Novelty:** 3
**Q2-2 Correctness-Technical Quality:** 3
**Q2-5 Clarity Of Writing:** 3

**Q1 Summary And Contributions:**

This paper studies adversarial MDPs where the cost functions may change over time. The authors develop a new policy search method that achieves a sublinear optimistic regret with high probability. This algorithm leverages the O-REPS method, incorporating a new biases cost estimator. The authors also introduce the anytime extensions for this problem by the doubling trick. The empirical evaluation validates the efficacy of the proposed algorithm.

**Q2-3 Extent To Which Claims Are Supported By Evidence:**

3: Good: the main claims are supported by convincing evidence (in the form of adequate experimental evaluation, proofs, (pseudo-)code, references, assumptions).

**Q2-4 Reproducibility:**

3: Good: key resources (e.g. proofs, code, data) are available and key details (e.g. proofs, experimental setup) are sufficiently well-described for competent researchers to confidently reproduce the main results.

**Q3 Main Strengths:**

1. Learning adversarial MDPs with bandit feedback is an important and well-motivated problem. This paper utilizes optimistic online learning methods to develop new algorithms for this setting.
2. The results in this paper are new and valuable to the community. The theoretical analyses are solid, though I don't check the proofs in detail.
3. The authors perform experiments to demonstrate the benefit of implicit exploration and cost predictors, which is valuable for a theoretical paper.
4. The paper is well-written and easy to understand.

**Q4 Main Weakness:**

1. The authors are encouraged to more distinctly emphasize the paper's novelty and contributions. The primary methods employed appear to be incremental and combinatorial, drawing on O-REPS as proposed by Zimin and Neu (2013), the optimistic loss estimator introduced by Wei and Luo (2018), and the concept of implicit exploration proposed by Neu (2015). Are there any innovative modifications or supplementary enhancements applied to these established methods? Highlighting such advancements would greatly clarify the paper's unique value and contribution.
2. This paper does not recognize several previous works that have achieved optimistic bounds for adversarial MDPs, including:

(ii) Fei et al. (2020) attained optimistic bounds for adversarial MDPs with unknown transitions through policy optimization methods.

(ii) Zhao et al. (2022) (full version on arXiv) obtained comparable optimistic bounds for adversarial MDPs. The techniques used in this paper are similar to theirs.

A detailed comparison with the works above is necessary.

[1] Fei et al. Dynamic Regret of Policy Optimization in Non-stationary Environments. In NeurIPS 2020.

[2] Zhao et al. Dynamic Regret of Online Markov Decision Processes. In ICML 2022. https://arxiv.org/abs/2208.12483

**Q5 Detailed Comments To The Authors:**

See the weakness part.

**Q9 Complying With Reviewing Instructions:**

Yes

---

> ### Author Rebuttal · Authors · 2024-04-07
>
> We thank you for your valuable feedback and for providing additional references. We summarized our response in two points.
>
> Novelty and contribution: While individual components of our algorithm have existed in prior literarture, our work is the first to propose and analyze the specific cost estimator featuring an optimistic predictor and implicit exploration within the context of AMDP. It is crucial to have the bounded variance and the optimistic regret bound at the same time to establish the theoretical and empirical results, including the new Bernstein-type inequality in Lemma 2 and the high-probability guarantee outlined in Theorem 3. Furthermore, our extensions to anytime and unknown transition settings represent non-trivial advancements, both algorithmically and theoretically. Finally, despite the absence of numerical experiments in existing online policy search algorithms for AMDPs, including Zimin and Neu (2013), Wei and Luo (2018), Lee et al. (2020), Jin et al. (2020) and Zhao et al. (2022), we verified our theoretical analysis via numerical experiment. Figure 1-(a) visually demonstrates our algorithm's superior convergence speed and reduced variance. In our final version of the paper, we will emphasize these points more clearly.
>
> Additional references:  First, Fei et al. (2020) studied policy optimization methods in AMDP with full information feedback and unknown transition function. Their algorithm estimates a state-action value function instead of a cost function to exponentially update the policy. Also, their second algorithm alternately updates policy and value function twice, mirroring the two-step optimization of Optimistic Mirror Descent. Conceptually, it is analogous to having a predictor as a Q-function that is updated with the previous episode's cost function. Since they analyzed dynamic regret, their bound is dependent on the magnitude of the optimal policies' difference between consecutive episodes. In the worst case, their regret bound scales as $O(T)$. In terms of static regret, where the comparison is against a fixed optimal policy and $P_T=0$, their bound scales as $O(\sqrt{T})$.
> Second, Zhao et al. (2022) investigated ensemble algorithms in the context of AMDP and more general settings, including episodic SSP and infinite-horizon MDP. In addition to the ensemble of occupancy measures, they imposed a lower bound on the occupancy measure for all states and actions. This regularization serves to bound the difference between the losses incurred by any two policies. They also analyzed the dynamic regret bound, which scales as $O(T\sqrt{\log(T)})$, and in terms of a static regret, it is $O(\sqrt{T})$.
> A pivotal distinction between both works and ours is that they exclusively explored the full information setting, a notably strong assumption that permits significantly simplified algorithm design and analysis. Furthermore, the regret bounds do not degrade with the estimation power of a predictor, and thus not optimistic.
> We will include these references to the final version of our paper.

---

### Meta-Review · Area_Chair_Hfqt · 2024-04-13

This paper investigates the optimistic regret bounds for adversarial Markov decision processes. The authors introduce new algorithms by tackling several technical difficulties. The reviewers unanimously agree that this work is of high quality. Congratulations to the authors for this nice work! The authors are also encouraged to take reviewers' feedback into account to enhance the clarity of the paper and improve discussions of prior works further.